# Microrobot collectives with reconfigurable morphologies, behaviors, and functions

Gaurav Gardi [1,2,8], Steven Ceron[3,8], Wendong Wang [4✉], Kirstin Petersen[5✉] & Metin Sitti [1,6,7✉]

Mobile microrobots, which can navigate, sense, and interact with their environment, could potentially revolutionize biomedicine and environmental remediation. Many self-organizing microrobotic collectives have been developed to overcome inherent limits in actuation, sensing, and manipulation of individual microrobots; however, reconfigurable collectives with robust transitions between behaviors are rare. Such systems that perform multiple functions are advantageous to operate in complex environments. Here, we present a versatile micro-robotic collective system capable of on-demand reconfiguration to adapt to and utilize their environments to perform various functions at the air–water interface. Our system exhibits diverse modes ranging from isotropic to anisotrpic behaviors and transitions between a globally driven and a novel self-propelling behavior. We show the transition between different modes in experiments and simulations, and demonstrate various functions, using the reconfigurability of our system to navigate, explore, and interact with the environment. Such versatile microrobot collectives with globally driven and self-propelled behaviors have great potential in future medical and environmental applications.

[1] Physical Intelligence Department, Max Planck Institute for Intelligent Systems, 70569 Stuttgart, Germany. [2] Department of Physics, University of Stuttgart, 70569 Stuttgart, Germany. [3] Sibley School of Mechanical and Aerospace Engineering, Cornell University, Ithaca, NY 14853, USA. [4] University of Michigan-Shanghai Jiao Tong University Joint Institute, Shanghai Jiao Tong University, Shanghai 200240, China. [5] Electrical and Computer Engineering, Cornell University, Ithaca, NY 14853, USA. [6] Institute for Biomedical Engineering, ETH Zurich, 8092 Zurich, Switzerland. [7] School of Medicine and College of Engineering, Koç University, 34450 Istanbul, Turkey. [8] These authors contributed equally: Gaurav Gardi, Steven Ceron. ✉email: wendong.wang@sjtu.edu.cn; kirstin@cornell.edu; sitti@is.mpg.de

Collectives in nature often make use of reconfiguration, altering the group's morphology to carry out complex functions in various environments[1–7]. At small scales, reconfiguration enables organisms to adapt to environmental disturbances and complete diverse functions. Inspired by the robustness and adaptability of these systems, engineers have mimicked naturally occurring behaviors through robot collectives that are programmable and interact with their environment to enable robust reconfiguration. For example, at the macro-scale, Kilobot collectives use programmed interactions to create different formations[8–10] and reconfigure to manipulate objects based on global signal inputs[11]. Other macro-scale robot collectives demonstrate how environmental interactions like contact-based coupling[12–15] can enable collectives to change their shape[16–18], function, and mode of locomotion[19–21]. At the micron-scale, reconfiguration in artificial systems is rare and it has the potential to open up possibilities in biomedicine, environmental remediation, and other applications[22,23]. At this scale, collectives interact through physical and chemical interactions to organize global responses beyond the reach of individuals[24–28]. The main paths for reconfiguration at the microscale include self-driven systems with active particles, and externally driven systems with particles controlled by one or more global signals[29]. In self-driven systems, local stimuli dominate a particle's behavior; particles react to their neighbor's actions to produce collective behaviors[30–34]. Externally driven microrobot collectives may be driven by magnetic fields[35,36], light[37,38], acoustics[39], or any other global stimulus that can alter the motion of one or more particles in the system. Given the high permeability of magnetic fields in many biological and non-biological materials used in biomedical applications, many researchers have focused on developing particle collectives[25] and soft materials[40,41] that are responsive to one or more quasistatic[42] or time-varying[43] magnetic fields. Therefore, a system that can utilize the mutual interactions between its constituents and respond to a global magnetic field stimuli to exhibit different behaviors would help realize a versatile reconfigurable robot collective.

One of the first magnetically actuated self-organizing systems was demonstrated by Grzybowski et al[44,45]., where millimeter-sized magnetic disks at the fluid-air interface spun in response to an external rotating magnet and assembled into hexatic patterns as a result of their hydrodynamic interactions. Manually changing the shapes of the magnetic particles[46] enabled the system to reconfigure by exploiting hydrodynamic interactions and a permanent magnet's potential well confinement to create self-organized collectives. Other systems, like the one used in this study, use capillary interactions by patterning edge corrugations on micro-disks so that the micro-disk collectives self-organize into square lattices[24] and rotating collectives[35,36]. Other microrobotic systems have demonstrated ribbon, chain or vortex formations[25–27,47–49], locomotion[42,50], and object manipulation[51,52]. One study shows impressive locomotion of two microrobotic swarms that can navigate through various mediums and complex environments; however, each swarm demonstrates a single mode, one creates ribbon-like formations while the other creates vortex-like formations[53]. Another experimental system produces four emergent modes (liquid, chains, vortices, and ribbons)[26], and it relies on a solid substrate for symmetry breaking to enable the formations and locomotion. Although this system exhibits four modes, the relative strength of different mutual interactions in the system is not clearly tunable; this limits the control of the collective's size, inter-particle separation, and each mode's local order. An ideal system should enable an external control parameter, like the magnetic field frequency, to dynamically program relative dominance among different particle–particle and particle–environment interactions to reconfigure between several tunable modes and their functions.

Moreover, a system that can transition on-demand from globally driven behaviors to a mutual interactions–dominated self-propelling behavior (like self-propelling Janus particles[30,31]) is yet to be realized. Such a system would not only be useful for robotics applications but also for fundamental studies to explore the link between globally driven and active systems.

Here, we present a micro-disk collective, consisting of around 120 micro-disks, at the fluid-air interface externally driven by time-varying magnetic fields that enable the collective to rotate, oscillate, remain static although individual micro-disks are dynamic, form chains, locomote through magnetic field gradients, perform contact-based and flow-based object translation and rotation, and explore an open space through gas-like self-propelling micro-disk pairs (Fig. 1). Each behavior is enabled by several particle-particle and particle-fluid interactions (e.g., hydrodynamic, capillary, and magnetic dipole–dipole interactions) that are controlled by external magnetic fields. Some of these behaviors are unique to our system and have no counterpart in other systems. Morover, our system shows both isotropic (rotating collectives) and anisotropic (chains) behaviors, and it also transitions from such globally driven behaviors to mutual interactions–dominated self-propelling behavior (like self-propelling Janus particles). The system's versatility and the micro-disks' ability to remain at the fluid-air interface is well-suited for practical applications like manipulating cells within a microfluidic chip, guiding development of micromachines, acting as a model system for exploring self-organization in soft matter, and studying collective behaviors that could finally translate to three dimensions and be used for microscale packaging[22] and medical applications, such as targeted active drug or other cargo delivery[54–56].

## Results

**Overview of collective behaviors**. All of the collective behavior modes in this study are produced by collectives of magnetic micro-disks at the air–water interface controlled by a biaxial oscillating uniform magnetic field (Supplementary Fig. 1). Micro-disks are driven by the global magnetic stimuli and locally interact through three pairwise forces: capillary, hydrodynamic, and magnetic dipole–dipole interactions. Each micro-disk is circular and has a six-fold symmetry given by six cosinuisoidal profiles along the disk's perimeter; the corrugations along each disk's boundary enables capillary interactions that can be attractive or repulsive depending on the relative disk orientations. The micro-disks' hexatic characteristic causes capillary interactions with six-fold symmetry. The number of symmetrically placed cosinusoidal profiles (between 2 and 6) would not significantly affect the behaviors presented in this work, however, the six-fold symmetry and circular shape are used because of simplicity in modeling the mutual interactions between such disks. When the corrugations on two micro-disks align, they attract; when they misalign above a threshold (~8°) they repel each other. A uniform ferromagnetic nanofilm of cobalt on each micro-disk's surface enables orientation-dependent magnetic dipole–dipole interactions. The hydrodynamic interactions are dependent on micro-disks' instantaneous spin speeds; faster spinning micro-disks exert greater hydrodynamic repulsion and create larger azimuthal flow fields that cause the collective to spread out. Our system differs from those reported earlier[35] in that it exploits two independent oscillating magnetic fields (Eq. 1), each along one of the axes on a horizontal plane, as compared to a rotating magnetic field used in the previous studies, to drive the ferromagnetic micro-disks. Therefore, the behaviors shown by the previous systems are a subset of those exhibited by the current system. Specifically, our system shows

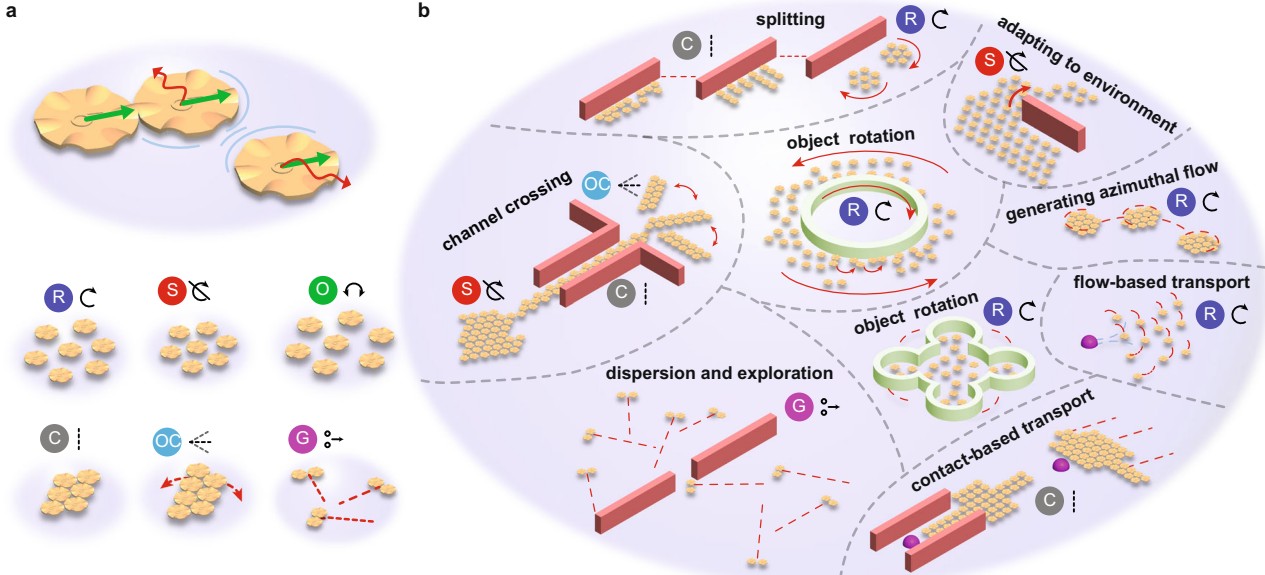

**Fig. 1 Microrobot collective reconfigurable behaviors and functions. a** Three micro-disks driven by external magnetic fields exhibit magnetic dipole–dipole attraction, capillary attraction, and hydrodynamic repulsion (top). Six collective modes are possible: rotation (R), oscillation (O), static (S), chains (C), oscillating chains (OC), and gas-like mode containing self-propelling pairs (G) (bottom). **b** Schematic of collective transitioning between the possible modes to perform various functions. Starting on the left side and continuing clockwise along the edge: the collective starts out in static mode and transitions to a chain to locomote through a narrow channel and exit the channel to form oscillating chains (channel crossing). At the top, the collective lines itself up against a wall and uses the physical interactions with the wall to separate into two clusters when it transitions to rotation mode (splitting). The collective can then pass around an obstacle more easily by adapting inter-disk distance through its static mode at high magnetic field frequencies (adapting to environment). The collective can then rotate and locomote at the same time (generating azimuthal flow) and induce motion on surrounding objects through its azimuthal flow field (object rotation and flow-based transport). At the bottom, the collective forms chains and pushes on an object (contact-based transport) and then disperses through a gas-like mode (dispersion and exploration). The center images show the collective can rotate objects through the azimuthal flow field when the micro-disks are within the object as well as encapsulating it.

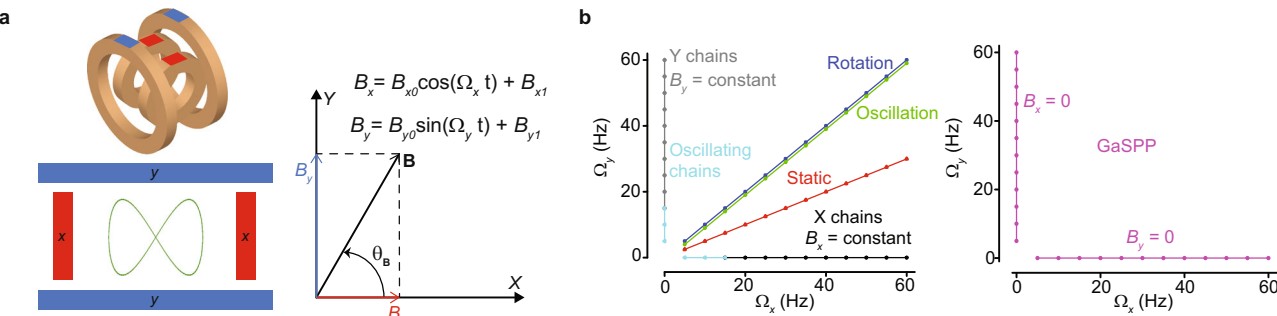

**Fig. 2 Collective modes within the frequency parameter space. a** The external magnetic field **B** modulated in 2D space by the axial magnetic oscillation frequencies, where $\theta_B$ is the orientation of **B** with respect to the x-axis. **b** Specific linear relationships between the two axial magnetic oscillation frequencies enables the different collective behaviors: Rotation (blue), static (red), oscillation (green), X chains (black), Y chains (gray), oscillating chains (cyan), gas-like mode containing self-propelling pairs, GaSPP (magenta). $\Omega_x$ and $\Omega_y$ are the oscillation frequencies of the external magnetic field along the x and y axes, respectively.

the same behaviors as the system reported earlier when the oscillation frequencies along both axes are equal,, i.e., $\Omega_y = \Omega_x$, which produces a rotating magnetic field. The use of two independent oscillating magnetic fields enables a richer set of behaviors, some of which are isotropic (e.g., hexatic-like circular collectives) while some are anisotropic (e.g., chains). Moreover, our system can transition from globally driven behaviors to mutual interactions-dominated self-propelling behavior and vice-versa. The magnetic field profiles are explained in Fig. 2, and the behaviors are shown and characterized in Figs. 3 and 4.

The external magnetic field exerts a torque on the micro-disks, trying to align the micro-disks with the external magnetic field

vector. The relative strength of the three pairwise interactions changes as the global magnetic field oscillates at different frequencies, which allows the collective to exhibit different modes and functionalities dependent on the relationship between the oscillation frequencies of the two orthogonal magnetic fields, $\Omega_x$ and $\Omega_y$, as shown in the line plots in Fig. 2 and the state map in Supplementary Fig. 2. Each colored line in Fig. 2b represents a different global behavior and is only one instance of several linear relationships that lead to that behavior. For instance, collective oscillation happens when the frequencies fulfill $\Omega_y = \Omega_x - 1$; however, the same general behavior could be found with $\Omega_y = \Omega_x - 2$ and $\Omega_y = \Omega_x - 3$. In this study, we do not exhaust all

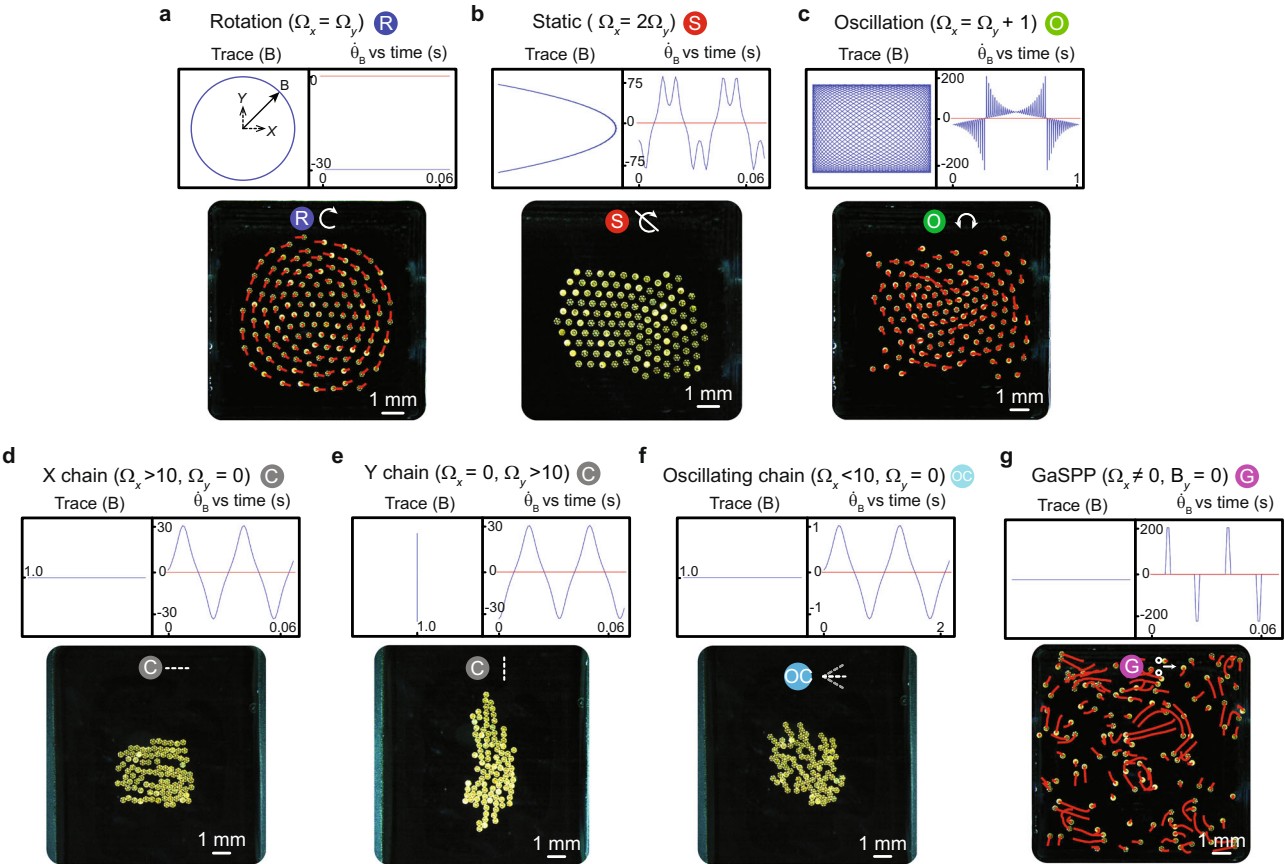

**Fig. 3 Demonstration of different collective modes. a–g** Summary of collective modes: Curve traced by the magnetic field vector Trace(B) (top left), the derivative of the orientation of magnetic field vector $\dot{\theta}_\mathbf{B}$ (blue) with respect to the x-axis (red) (top right), representative experimental images of the collective exhibiting respective behaviors (bottom). The brightness of the images is enhanced using photoshop for better visualization. The symbols on top of the experimental images in (**a–g**) represent the mode that the collective exhibits. These symbols correspond to those shown in Fig. 1a. $\Omega_x$ and $\Omega_y$ are the oscillation frequencies of the external magnetic field along the x and y axes, respectively.

possible linear combinations that lead to particular behaviors, but instead focus on several distinct modes that enable useful functions and examine the collective's properties at one linear relationship for each mode. The magnetic field vector **B** is defined as:

$$\mathbf{B} = \begin{bmatrix} B_{x0} \cos(2\pi\Omega_x t) + B_{x1} \\ B_{y0} \sin(2\pi\Omega_y t) + B_{y1} \end{bmatrix},$$ (1)

where $B_{x_0}$ and $B_{y_0}$ are the amplitudes of the axial time-varying fields (10 mT), $B_{x_1}$ and $B_{y_1}$ are the constant axial fields ($B_{x_1}$ and $B_{y_1}$ are 0 for all modes except the X-chains where $(B_{x_1}, B_{y_1}) = (0, 10)$ mT), and t is time. As illustrated at the top of each subfigure in Fig. 3, the axial frequencies affect the trace of **B** along the two-dimensional (2D) workspace and the time-dependent behavior of $\dot{\theta}_\mathbf{B}$; **B** drives the orientation of each micro-disk as it aligns its magnetic dipole axis with the external field, and $\dot{\theta}_\mathbf{B}$ indicates the rate at which the external field changes its orientation. For example, $\Omega_y = \Omega_x = \Omega$ produces a rotating magnetic field, the trace of **B** is a circle and $\dot{\theta}_\mathbf{B}$ is constant and equal to $\Omega$. This rotating magnetic field exerts a torque on the micro-disks such that each disk spins about its own center of mass. The linear relationship between $\Omega_x$ and $\Omega_y$ changes each micro-disk's angular velocity and its axial oscillation over time. These variations along with the capillary, magnetic, and hydrodynamic interactions enable the global formations and functions described in the following sections (Supplementary Movie 1).

**Rotation mode**. For $\Omega_y = \Omega_x$ (blue line in Fig. 2b), **B** traces a circle, resulting in a rotating magnetic field that exerts a torque on the micro-disks (Fig. 3a (top)). Consequently, each micro-disk spins about its own axis at the same angular velocity as the external magnetic field, below the step-out frequency (~65–75 Hz), where the external magnetic torque is insufficient for synchronized disk rotation. Each spinning micro-disk creates an azimuthal flow field, which enables the collective to orbit around a common center of mass. Each disk's spin speed and the inter-disk hydrodynamic repulsion increases with frequency while the local hexatic order parameter decreases. This increased repulsion causes the collective to spread out and orbit more slowly (Fig. 4a–c). Also, the micro-disks further away from the collective's center of mass revolve faster than those closer to it (the red trajectory lines in the image in Fig. 3a) (see Methods subsection on experimental protocol).

We reproduce the rotation behavior in simulations using a numerical model constructed using the pairwise interactions (Supplementary Fig. 3). The agreement between the experiments and simulations suggests that the numerical model contains the right ingredients responsible for the behaviors of our system (for more details see the Methods subsection model for simulations). Above 60 Hz, the collective expands and conforms to the physical boundary's shape, however, the simulations retain the collective's circular shape across all frequencies. Although this affects the experimental evaluation of the collective radius, it exemplifies how confinement forces the collective to adapt its morphology. This mode's remarkable

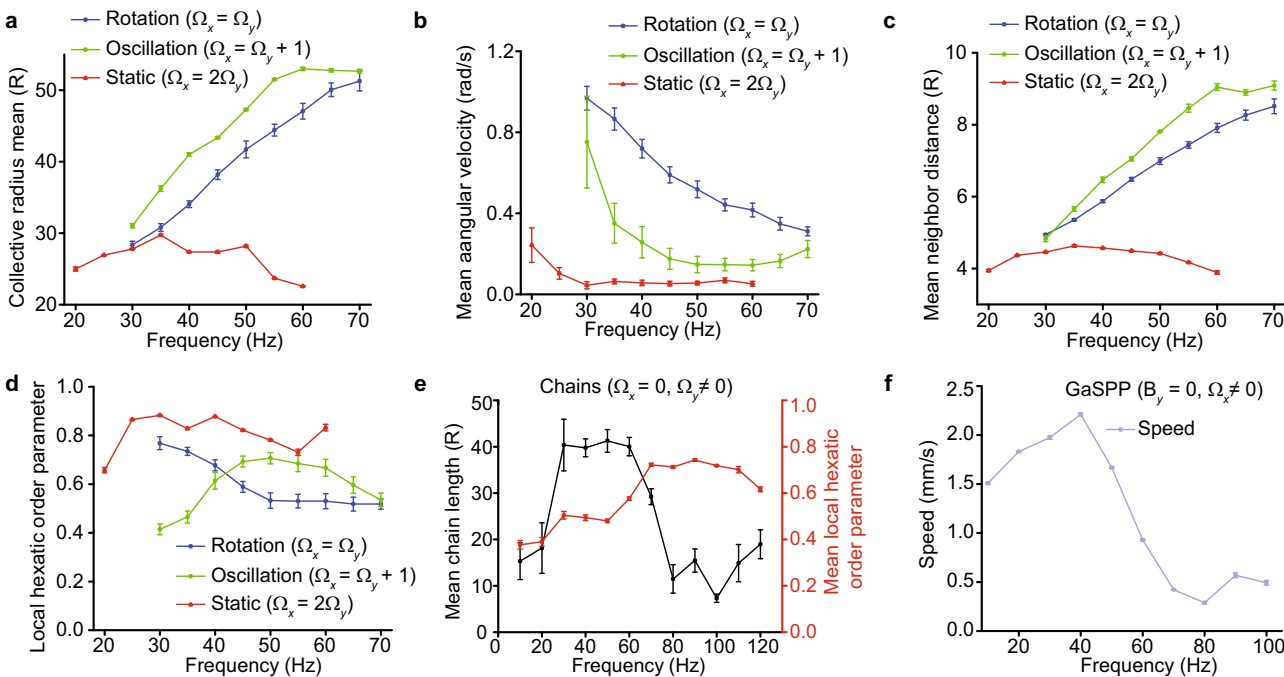

**Fig. 4 Characterization plots for the microrobot collectives of 120 micro-disks in different modes. a** Collective radius, **b** mean angular velocity, **c** mean neighbor spacing, and **d** mean local hexatic order parameter for the rotation (blue), static (red) and oscillation (green) modes. **e** Mean chain length (black) and mean local hexatic order parameter (red) for the chain mode. **f** Mean speed for the GaSPP mode. Error bars in (**a–f**) represent standard deviations over 10 s.

feature is the collective azimuthal flow field, which is used for various functions as demonstrated later.

**Static mode.** The static mode is a novel behavior composed of individually dynamic micro-disks where each disk's center of mass remains static with respect to the global reference frame, but the disk oscillates about its own center axis. This mode is characterized for $\Omega_x$ and $\Omega_y$ lying on the $\Omega_y = \frac{\Omega_x}{2}$ line (red line in Fig. 2b and Supplementary Fig. 2). This linear relationship between $\Omega_x$ and $\Omega_y$ produces a parabolic trace of **B**, which enables a time-varying profile for $\dot{\theta}_\mathbf{B}$ (Fig. 3b). The oscillations of $\dot{\theta}_\mathbf{B}$ across the x-axis indicate that micro-disks change their direction of oscillation and angular velocity over time, oscillating while fixed in place and interacting with their neighbors via time-varying pairwise interactions, which cancels out any fluid motion in the short term and keeps the collective static. Increasing the oscillation frequency increases micro-disks' effective hydrodynamic repulsion, which increases their mean neighbor separation distance until the disks start to step out around $\Omega_y = 50$ Hz (Fig. 4c). The distance between neighbors within the collective allows it to encapsulate objects while maintaining a high local and global hexatic order across all frequencies (Fig. 4d and Supplementary Fig. 4). Regardless of the object's size, the collective remains cohesive much more easily in this mode; in contrast, during the collective rotation, all micro-disks exhibit higher hydrodynamic repulsion on each other, which enables dispersion in the presence of non-uniform external forces like repulsion from boundaries and objects.

**Oscillation mode.** Oscillating collectives are characterized for $\Omega_y = \Omega_x - 1$, with frequencies in the same range as rotating collectives. For this mode, $\dot{\theta}_\mathbf{B}$ oscillates for several periods either below or above the x-axis (Fig. 3c), meaning disks keep a constant spin direction in the short term but with angular speed

fluctuations. This driving signal enables the whole collective to periodically switch between clockwise and counter-clockwise rotations with a time-varying angular speed. The key difference between the static and oscillation modes is that **B** has a much longer period for the oscillation mode. This causes uniform collective rotation in one direction for short-time observation, while the mean long-time angular speed of this mode is still smaller than that of the rotation mode. The characterization experiments show that the neighbor separation distance increases with higher frequencies, and the average angular speed of the oscillating collective is smaller than that of the rotating collectives (Fig. 4a–c). The local hexatic order for this mode is lower than for the static mode, but higher than the rotation mode. This mode enables the collective to remain robust against unfavorable system conditions like defective particle corrugations or irregularities in the fluid medium and surrounding boundaries.

**Chain mode.** Magnetic fields oscillating about a fixed mean-axis of oscillation create chain formations aligned along the x or y axis, or any vector between the two. Chains along the x or y axis only require a frequency along the alignment axis while the other axis frequency is zero (Fig. 3d–f). Chains along any direction in 2D can be formed as shown in Supplementary Fig. 2b (see Methods subsection on experimental protocol for more details).

Throughout the experimental characterizations and demonstrations, the chains were kept either in the x or y direction. At low frequencies ($\Omega_{x,y} < 50$ Hz) the attractive forces surpass the repulsive ones and the micro-disks align and attach along edge corrugations forming connected chains. For even lower frequencies ($\Omega_{x,y} < 10$ Hz), the collective oscillates about its center of mass while following **B**, resulting in oscillating chains. At higher frequencies ($\Omega_{x,y} > 50$ Hz), the micro-disks separate due to greater hydrodynamic repulsion, thereby forming chains with separated micro-disks, whose neighbor separation distance increases with frequency (Fig. 4e). At these higher frequencies,

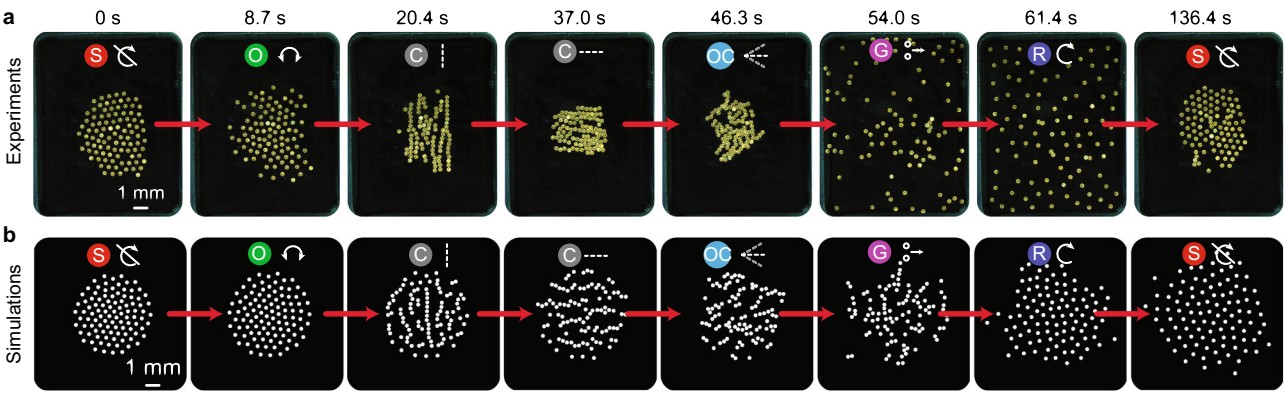

**Fig. 5 Collective mode transitions between different collective formations defined by the values for $\Omega_x$ and $\Omega_y$. a** Experimental transitions (from left to right): static ($\Omega_x = 60$ Hz, $\Omega_y = 30$ Hz), oscillation ($\Omega_x = 31$ Hz, $\Omega_y = 30$ Hz), Y chains ($\Omega_x = 0$ Hz, $\Omega_y = 30$ Hz), X chains ($\Omega_x = 30$ Hz, $\Omega_y = 0$ Hz, $B_y = 10$ mT), oscillating chains ($\Omega_x = 1$ Hz, $\Omega_y = 0$ Hz, $B_y = 10$ mT), GaSSP ($\Omega_x = 70$ Hz, $\Omega_y = 0$ Hz, $B_y = 0$ mT), Rotation ($\Omega_x, \Omega_y = 70$ Hz), Static ($\Omega_x = 60$ Hz, $\Omega_y = 30$ Hz). **b** Simulated mode transitions corresponding to the experiments in (**a**). The last frame of a mode in the simulations is used as the first frame in the simulation of a subsequent mode. The brightness of the images in (**a**, **b**) is enhanced using photoshop for better visualization. The symbols on top of the sub-images in (**a**, **b**) represent the mode that collective exhibits. These symbols correspond to those shown in Fig. 1a.

the collective resembles formations given by similar frequencies in the static mode with high hexatic order (Fig. 4e). Although the two modes seem similar in spatial configuration, the orientation of the chains can be controlled more reliably than that of the static configuration.

It is worth noting that although intuitively it seems that a static magnetic field could be used to create chains, this system indeed requires an oscillating magnetic field. The chain formations are formed as a result of the synchronous orientation oscillation of micro-disks throughout the collective, the flow generated by each micro-disk, and its effect on neighboring constituents. If a static magnetic field is applied, the micro-disks would cease to oscillate about their axes and the capillary interactions would dominate, which would result in formation of hexagonal clusters.

**Gas-like self-propelling pairs (GaSPP) mode**. One-dimensional (1D) oscillating magnetic fields (like in eq. 2) produce a gas-like formation in which the collective breaks into several micro-disk pairs that randomly disperse, similar to the way active self-propelling Janus particles behave when they are stimulated[30,31]. The key difference here is that the micro-disks' collective behaviors are determined by controllable global stimuli, whereas active Janus particles are mainly driven by local stimuli. The micro-disk pairs translate perpendicularly to the line joining their two respective centers (Fig. 3g); their translation speed increases with frequency and drops to zero as soon as the pair collides with a boundary or other micro-disks, both points at which the pair permanently or temporarily break apart. The micro-disks translate in a new direction if they rejoin or become pairs with other micro-disks. Interestingly, two pairs frequently collide and exchange partners before moving off in new directions. Tight clusters and single micro-disks far away from others remain at a location until other micro-disk pairs collide with them. The average velocity of the pairs in this mode increases with frequency until a threshold (~50 Hz) when the collisions of the pairs with the physical boundary becomes more frequent, thereby reducing their average velocity (Fig. 4f). This mode enables collectives to disperse and fill open areas at different speeds. An example of the magnetic field profile that enables GaSPP mode is the following:

$$\mathbf{B} = \begin{bmatrix} 10\cos(40t) \\ 0 \end{bmatrix}. \qquad (2)$$

When two disks spinning in opposite directions come close, the net azimuthal flow created by them causes them to translate

as a pair, just in the same way as when two disks spinning in the same direction (as in the rotation mode), start orbiting around their common center of mass. When the external 1D oscillating magnetic field that enables GaSPP, changes its direction, it makes an angle of 180° with the magnetic dipole on the disks, creating an unstable equilibrium. Because of this unstable equilibrium, the micro-disks can either spin clockwise or counter-clockwise with equal probability, to align their magnetic dipole with the external magnetic field. Thus, the collective breaks into pairs composed of one disk each that spins in either direction. These pairs translate in random directions and their translation speed can be controlled by the frequency of the oscillating magnetic field.

The GaSPP mode opens up possibilities for several fundamental studies investigating the systems of active particles. For example, studies like those investigating the emergence of directed motions in a collective of active particles[57], could also be possible using the GaSPP mode, by tuning the density of the micro-disks in GaSPP mode (the speed of the pairs can be tuned using the external magnetic field) to study the emergence phenomenon in such systems. The GaSPP mode could be advantageous over other systems, like the quincke rollers, or the active janus particles, because the mutual interactions among the micro-disks in GaSPP mode are relatively simpler than the non-trivial fluidic interactions among the quincle rollers and also GaSPP does not require any fuel (as is the case for the chemotactic janus particles). Moreover, the speed of the pairs in the GaSPP can be tuned by the external magnetic field. The GaSPP behavior could also be useful for demonstrating theories such as the long-range order in active systems. The GaSPP mode can be useful for robotics studies as well. For example, the GaSPP can be used as the microrobot swarm to create superstructures like the ones developed using the rodlike vibrating robots[58,59].

Supplementary Figs. 5 and 6 shows that the collectives exhibit all the above mentioned behaviors even with a low number of micro-disks.

**Transitions between different modes**. The mode transitions are demonstrated via experiments and simulations (Fig. 5, Supplementary Fig. 2c and Supplementary Movies 2–5). A rotating collective of around 120 micro-disks starts out at $\Omega_{x,y} = 70$ Hz and transitions to a rotation mode at $\Omega_{x,y} = 20$ Hz, this increases the collective's average angular velocity and decreases its size. The collective then transitions from the rotation mode to a static

collective, followed by a transition to the oscillation mode. Further, the system transitions from an isotropic oscillation mode to anisotroic Y chains. The Y chains transition to X chains by changing the alignment axis; the collective then switches to oscillating chains, where each chain oscillates about the x-axis. Thereafter, the collective transitions from the globally driven chains to a self-propelling mode, where the micro-disks disperse through the GaSPP mode and transition to rotation mode at higher frequencies. Finally, the static mode collapses the collective back to the center and forms a stationary hexatic-like structure. The transition sequence was simulated using a numerical model where the collective starts with rotation mode ($\Omega_{x,y} = 20$ Hz) and transitions through the whole sequence and ends with the static mode (Fig. 5b and Supplementary Movie 3). The qualitative agreement between the experiments and simulations indicates that our system's behaviors can be explained using the main pairwise interactions between the micro-disks and external magnetic fields (for more details on the numerical model see Methods subsection on model for simulations and model for GaSPP simulations). Note that there are some discrepancies between experiments and simulations in the final formations for the rotation mode ($\Omega_{x,y} = 70$ Hz) and static mode ($\Omega_x = 60$ Hz, $\Omega_y = 70$ Hz). These discrepancies arise because we do not consider the hydrodynamic drag from the arena boundary, which remains insignificant in most cases but becomes significant at higher frequencies of the external magnetic field. Because the modeling of the drag from the arena boundary is highly non-trivial and because the absence of this term does not influence the simulation results significantly, this term was not included in our numerical model. Additionally, a concave air–water interface can drive the disks towards the center of the arena, creating an effective attraction between the disks. The simulations assume a flat air–water interface and each experiment approximates this; however, even a small concavity in the interface lowers neighbor spacing between micro-disks in the experiments, constraining the motion of individual disks. This difference is most clearly evident in the simulations of the chain mode.

**Mode-enabled robotic functionalities**. Next, we demonstrate diverse functions using different collective modes and quantitatively characterize several key parameters, including locomotion and dispersion speed under various external field parameters and angular velocities of the manipulated objects. These demonstrations begin with magnetic field gradient-based mechanisms that enable collective locomotion and navigation through complex environments and enable contact-based object transport (Fig. 6 and Supplementary Movie 6). Figures 7, 8 (and Supplementary Movies 7–14) demonstrate the collective's ability to manipulate itself and the surrounding environment through the fluid medium; this includes flow-induced locomotion, object transport, object rotation and orientation control. Finally, we show the useful functions available through the GaSPP mode: dispersion and space-filling with directional flows, and splitting by using the environment boundaries (Fig. 9 and Supplementary Movies 15–17).

We use 2D external magnetic field gradients to enable collective locomotion. We characterize around 120 micro-disks' average locomotion speed as a function of their collective mode, field gradient along the x and y axes, and field frequency (Fig. 6a, b and Supplementary Movie 6) and collective size (Supplementary Figs. 7, 8). The chain mode with alignment along the gradient's axis exhibits the highest speed among the four tested modes. Note that the chains slow down when they approach the arena's boundary as a result of the confinement effect; this results in higher standard deviations (Fig. 6a–c).

The collective moves on desired trajectories to form the letters MPI and C, while maintaining a chain formation (Fig. 6d and Supplementary Movie 7). Next, we demonstrate the system's ability to switch between collective modes that enable it to navigate through a complex environment (Fig. 6e and Supplementary Movie 8). Throughout this environment, the collective passes through narrow passages that require it to switch between X- or Y-chains in order to keep all micro-disks together. Additionally, the collective moves quickest as a chain, when the chain is aligned along the direction of the magnetic field gradient. After about 150 s, we switch the collective to the GaSPP mode, demonstrating that the collective can disperse within a complex arena and coalesce afterwards through the rotation mode.

We also show object transport using physical contact (Fig. 6f and Supplementary Movie 9) through a collective that must navigate towards and around an object and then reconfigure to push it in the desired direction. The collective starts out at the arena's left side in static mode and passes the spherical object (1 mm-diameter polystyrene bead) as X-chains, driven by a magnetic field gradient (0.7 Gauss/mm). It is important that the collective exhibits the X-chain mode when moving to the right side of the arena since this allows it to maximize its speed and minimize the contact surface area to prevent pushing the bead to the right side. Then the collective switches to Y-chains and pushes the bead to move left. The Y-chains ensure that the collective has a large surface contact area along which it can push the object from the right side of the arena to the left.

Flow-induced locomotion enables a collective to move around by taking advantage of the generated fluid flows and surrounding environmental boundaries. When the collective reaches the arena's bottom right corner (after 190 s in Fig. 6e) it transitions into a clockwise rotation mode with a constant downward field gradient and moves towards the left, opposite to how a wheel rolls on a boundary. We speculate that this counter-intuitive leftward movement is caused by a combination of the clockwise flow and the symmetry breaking due to the presence of a fixed boundary on one side of the collective. A detailed study will give insight into the specific mechanisms that enable this collective behavior; however, this is beyond the scope of this paper. The complex interplay between parameters like magnetic field gradient, boundary conditions, and biaxial magnetic field frequencies opens up many avenues for researchers to explore optimal control methods for transporting a collective within complex and/or dynamic environments.

The emergence of flow-induced locomotion results from the azimuthal flow generated by each spinning micro-disk in rotation mode. Figure 7a shows a fluorescent dye that is introduced to the left of a clockwise-rotating collective and spreads through a combination of its own diffusion and the azimuthal flows generated by the collective. A close look at the images shows that the dye circles around individual micro-disks' edges and is swept by azimuthal flows onto other nearby micro-disks (Supplementary Movie 10).

Contact-free object transport is achieved through the rotation mode. First, a collective of 120 micro-disks rotates to move a polystyrene ball (1 mm) from the top center of the arena to the lower right corner (Fig. 7b). The collective rotates in the counter-clockwise direction to generate a flow that drags the ball along, thus transporting it into the greater open space in the lower right corner of the arena. Similarly, the ball can be transported to the arena's left side using a clockwise rotation (Supplementary Fig. 9 and Supplementary Movie 11).

The second demonstration of contact-free object transport is shown in Fig. 7c where the goal is to move the ball to the arena's bottom right corner. Similarly to the experiments in Fig. 6f, the collective navigates to the right side of the arena and around the

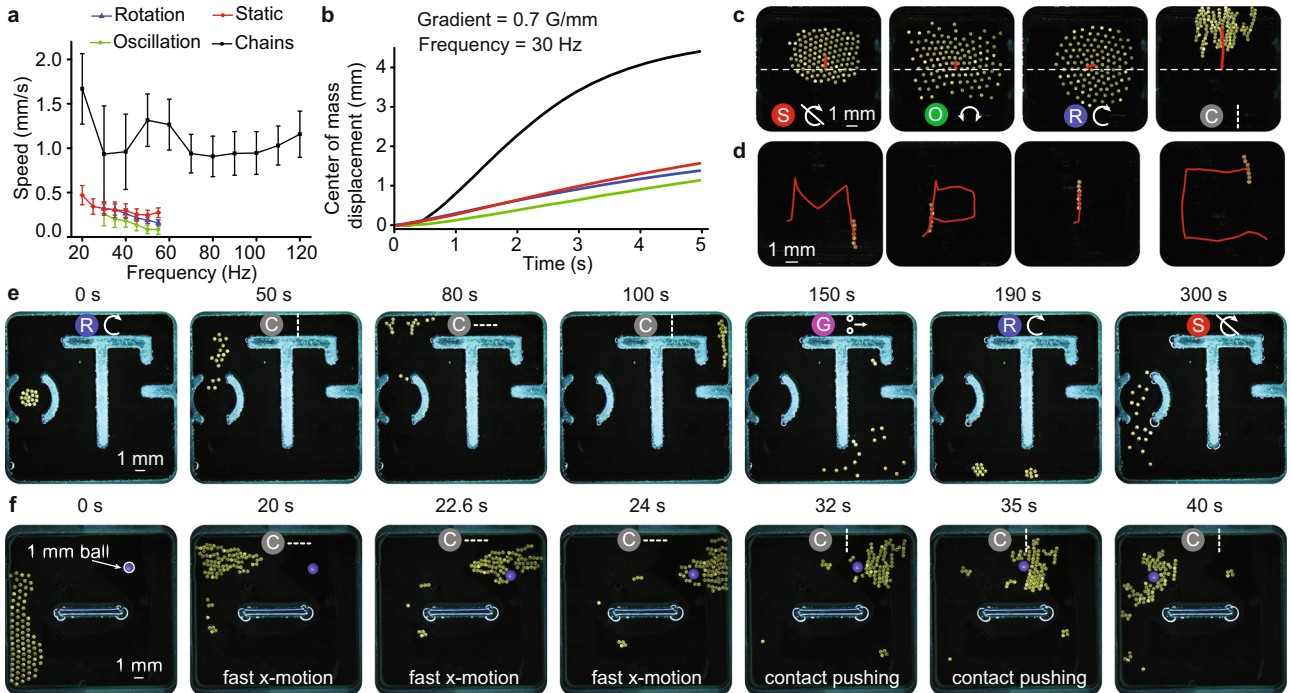

**Fig. 6 Magnetic gradient-assisted collective navigation through confined environments and contact-based object transport. a** Locomotion speed comparison for the rotation (blue), static (red), oscillation (green), and chain (black) modes with $N = 126$ micro-disks driven at different frequencies when the gradient is $F = 0.7$ Gauss/mm. The chains move faster than other modes. The error bars represent standard deviations over 5 s. **b** Displacement of each mode over time when at $\Omega_y = 30$ Hz. The chains move faster and get slowed down due to the physical boundary around 2.5 s. This contributes to large standard deviations in chain speed in (**a**). **c** Representative images showing the trajectories of static (left), oscillating (middle left), rotating (middle right), and chain (right) collectives under the influence of a magnetic field gradient along the y axis. **d** Y chain of seven micro-disks driven with magnetic field gradients to produce MPI and C trajectories (Supplementary Movie 7). **e** 17 micro-disks transition between different modes to navigate through narrow passages (Supplementary Movie 8). (**f**) The collective switches between the modes static, X chains, and Y chains to locomote and push an object. The brightness of the images in (**c**–**f**) is enhanced using photoshop for better visualization. The symbols on the sub-images at the bottom, in (**c**), and top, in (**e**, **f**), represent the mode the collective exhibits. These symbols correspond to those shown in Fig. 1a. The sub-images in (**f**) are labeled at the bottom with the function that the collective performs.

bead. As opposed to the demonstration in Fig. 6f where the collective must be aligned adjacent to the bead to transport it, here the collective must encapsulate the bead so there is a significant density of micro-disks around it in all directions. After locomoting to the arena's right side at a high speed through X-chains, the collective switches to static mode so that micro-disks can evenly distribute throughout the bead's perimeter without transferring torque that might send the bead in an undesired direction. The micro-disks begin to spin clockwise to generate a flow that moves the ball towards the left. Because the arena has a center barrier with fluid all around it (rectangular loop design), the ball makes its way to the left side, downward, and then back to the right side where it finishes below its starting point. The cumulative effects of the boundary, the size of the collective, their direction of rotation, and the general arena design dictate the ball's trajectory. This experiment demonstrates the capability of our system to use external factors, like boundaries, to achieve otherwise difficult tasks. Although it is beyond the scope of this paper, the complex interplay between these different parameters offers great opportunities to develop robust, efficient, and real-time control strategies for targeted manipulation of one or multiple objects.

We have shown the collective can transport large spherical objects through contact-based and contact-free mechanisms; however, it is also useful if these collectives control single or multiple objects that encapsulate the collective, or that the collective itself encapsulates. Figure 8 demonstrates the various flow-induced object manipulation behaviors possible.

The azimuthal flow fields from the individually spinning micro-disks, placed inside a ring, creates a larger circular flow around the collective's perimeter; the viscous drag on a ring's inner boundary enables it to rotate (Fig. 8a). Figure 8b shows that the collective can also control the motion of an object externally, as would be needed for solid objects. In these cases, a ring's direction of rotation is opposite the individual micro-disks' spin direction. The strength of the azimuthal flows generated by the collective can be changed in three ways: by using different numbers of micro-disks (varying the area density), by using rings of different sizes while keeping the number of micro-disks constant, and by changing the rotation speed of the collective. These experiments were performed for three ring objects of different sizes and with different numbers of micro-disks inside or outside the structure in each case. Their characterization demonstrates that increasing the frequency of the external magnetic field decreases the collective's angular speed (Fig. 8a–c and Supplementary Figs. 10, 11). Interestingly, at low frequencies several micro-disks form pairs and roll along the outer perimeter of the rings in several instances (Supplementary Movie 12). The rotational manipulation is not restricted to objects of circular shapes; Fig. 8d, e show rod- and star-like shapes encapsulating collectives, rotating about their center axis as a result of the azimuthal flows.

The next set of demonstrations address dual object manipulation when there are collectives within one or both of the objects (Fig. 8f, g). Two rings of different sizes, each encapsulating different numbers of micro-disks (34 in a small and 60 in a large

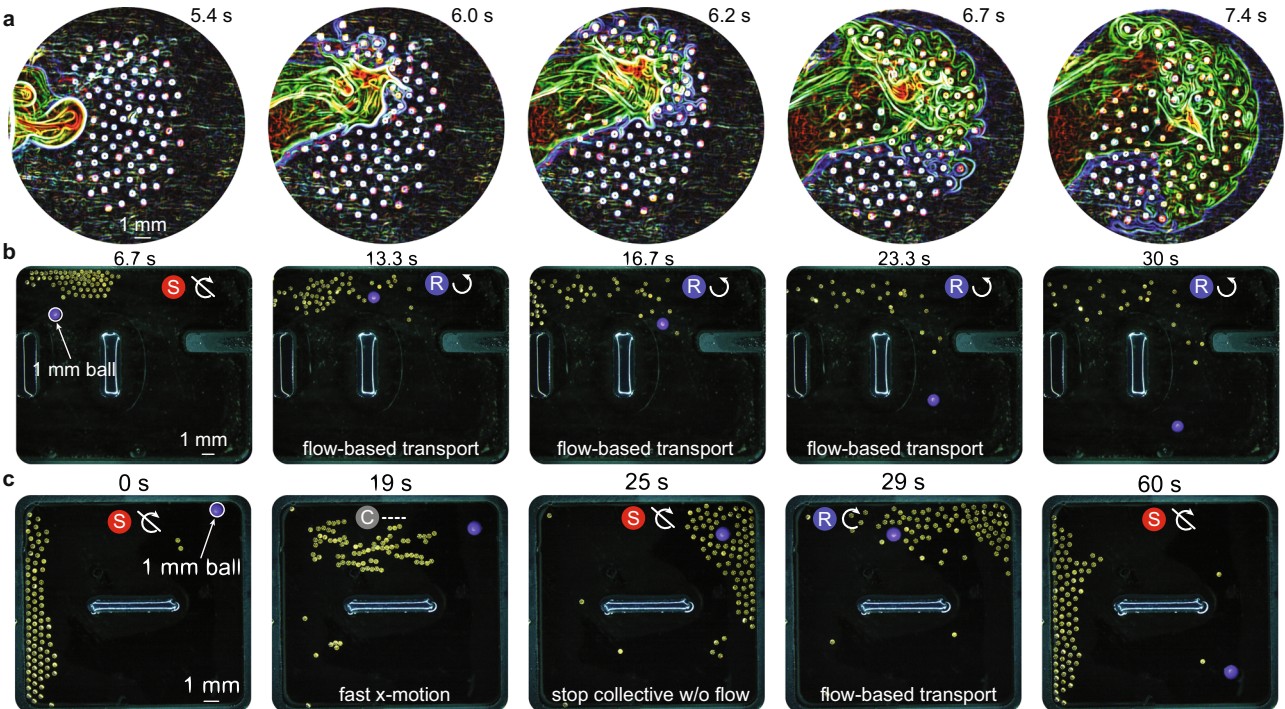

**Fig. 7 Flow-assisted contact-free object transport. a** Qualitative demonstration of flow around micro-disks when the collective is rotating. Green and blue lines were extracted using edge detection (see Supplementary Movie 10 for the unprocessed video). **b** Collective rotates to guide a purple ball to the bottom right-hand corner. **c** Collective rotation guides a ball around the perimeter of the arena by using the arena design. The brightness of the images in (**a–c**) is enhanced using photoshop for better visualization. The symbols on top of the sub-images in (**b, c**) represent the mode the collective exhibits. These symbols correspond to those shown in Fig. 1a. The sub-images in (**b, c**) are labeled at the bottom with the function that the collective performs.

ring, respectively), rotate in the same direction, but since they touch each other's boundary, their motion becomes coupled due to capillary torque and the two rotate about their common center of mass. However, when only one of the rings encapsulates a collective, the torque is exerted from the driven ring to the passive one, like gears, and as a result they rotate in opposite directions (Fig. 8g). This way the collective is used to demonstrate torque transfer between two rings, one ring actuated by the collective, while the other ring rotates either indirectly about its own axis or about a common center of mass with the first ring (Supplementary Movie 13).

Using the experiments discussed above, we finally demonstrate that the collective can switch between different modes to manipulate a C-shaped object with a small opening, stop it at a desired orientation to enter it, rotate it from the inside, and then exit (Fig. 8h and Supplementary Movie 14).

Finally, we show that the propulsion behavior in the GaSPP mode can be useful to disperse the collective (Fig. 9a). More specifically, this mode enables small collectives to disperse across large areas, within an object without transferring a significant amount of torque, as well as escape a free-floating structure without changing its orientation. We show that the mean velocity of the pairs increases with oscillation frequency ($\Omega_x$) of the 1D magnetic field (Fig. 4f). We compare the dispersion of micro-disks when using the expanded rotating collective and the GaSPP mode (Fig. 9b–c and Supplementary Movie 15). In Fig. 9b, we place the rotating collective ($\Omega_{x,y} = 20$ Hz) inside an arbitrary shape and then transition to the rotating collective at $\Omega_{x,y} = 70$ Hz, and in Fig. 9c the collective transitions to a GaSPP mode at $\Omega_x = 70$ Hz. We find that a rotating collective rotates the whole boundary while uniformly spreading across the whole available area, whereas the collective in GaSPP mode expands less uniformly, much faster, and without rotating the boundary.

Finally, we demonstrate two different ways to split up the collective. We first show the temporary splitting of the collective using reconfiguration into other modes and the physical boundary (Fig. 9d, Supplementary Fig. 12 and Supplementary Movie 16). In this case, the collectives split into two circular, rotating collectives for a short time until the inter-disk attraction takes over and they merge into a single collective. Then we demonstrate the use of GaSPP to split the collective and then merge the two small collectives using the chains, aided by magnetic field gradients, to navigate through a narrow opening (Fig. 9e and Supplementary Movie 17). Specifically, we place the micro-disks inside a rectangular arena that is divided into two squares by a wall. The only way for the micro-disks to go from one side of the arena to the other is via a small (~1 mm) opening at the center of the dividing wall. We start by placing the collective in the left square, forming Y-chains, and then changing to GaSPP mode ($\Omega_x = 70$ Hz). This mode enables fast dispersal of the micro-disks to the right side of the arena and on switching to the rotation mode ($\Omega_{x,y} = 50$ Hz), the collective forms two rotating groups, one on each side of the arena. The two groups then merge by transporting the group on the right side using the X-chains to go back to the left side through the narrow opening at the center, when assisted by a magnetic field gradient. This experiment highlights the capability of our system to adapt to and explore diverse environments by reconfiguring into different formations. Aside from an application perspective, this novel mode has great potential in carrying out fundamental studies on self-propelling particles; although the behavior is enabled by mutual interactions between particle pairs, their speed is tuneable through a global magnetic field frequency. Studying the fundamental behaviors of this mode more closely could give insight into the possibilities of self-propelling collectives like those composed of Janus particles.

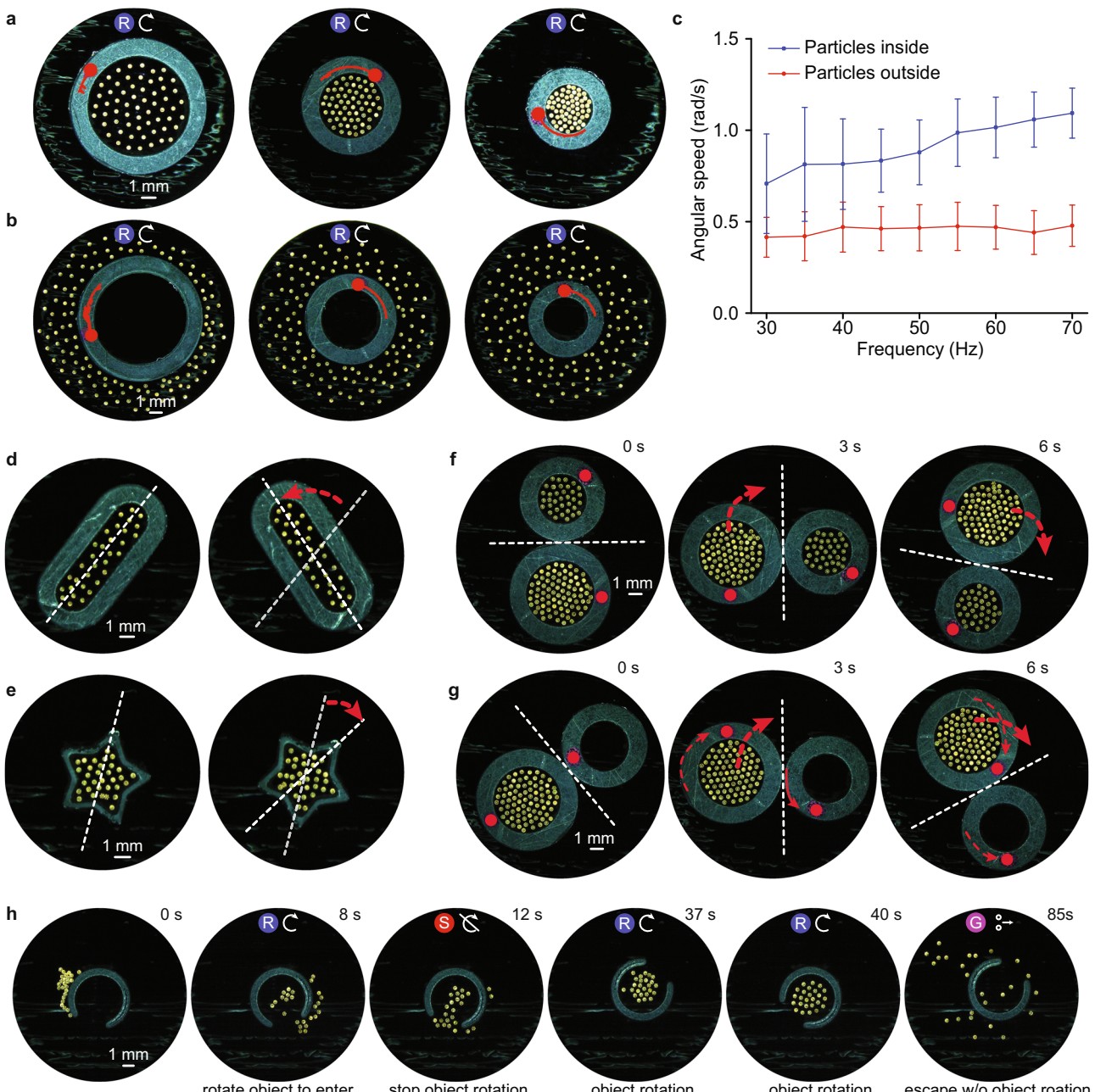

**Fig. 8 Flow-assisted object rotation and orientation control. a** Micro-disk collectives within ring-shaped objects of different sizes; the object rotates in the same direction as the collective. **b** Collectives encapsulate and rotate ring-shaped objects; the object rotates in the opposite direction as the collective. Red line in (**a**, **b**) shows the rotational trajectory of the rings. The same rings are used for (**a**), (**b**). **c** Angular velocity of a ring-like object (ID 3.4 mm and OD 6.5 mm) when there are 60 micro-disks inside or outside the structure. Error bars represent standard deviations over 10 s. **d** Collective rotates a rod-shaped object and **e** a star-shaped object. **f** Two collectives within adjacent objects enable coupled object rotation. **g** A collective within one of two adjacent ring-like objects rotates the structure it is in and exerts a torque on the neighboring object to enable gear-like motion. **h** A collective enters a C-shaped object, rotates it, and exits the structure. A red circle is added to highlight the orientation of the rings (**a**, **b**) and (**g**, **f**). The red dashed arrows in (**d**–**g**) show the direction of rotation of the individual objects (thin arrows) and their center of mass (thick arrows). The brightness of the images in (**a**, **b**) and (**d**–**h**) is enhanced using photoshop for better visualization. The symbols on top of the sub-images in (**a**, **b**) and (**h**) represent the mode that the collective exhibits. These symbols correspond to those shown in Fig. 1a. The sub-images in (**h**) are labeled at the bottom with the function that the collective performs.

## Discussion

In summary, we present a microrobot collective system that can reconfigure into six different behaviors, each of which is capable of diverse tasks. First, we show the different behaviors our system exhibits, both via experiments and simulations. These behaviors include globally rotating, static and oscillating formations composed of dynamic micro-disks, static and oscillatory chains, and a novel gas-like mode composed of self-propelling pairs. Our system exhibits various formations, each possessesing further tunability of collective size, angular velocity, and locomotion speed. We further characterize each behavior at different frequencies and demonstrate inter-mode transitions. Finally, by exploiting the reconfiguration capabilities of our system, we realize several functions like collective navigation through confined

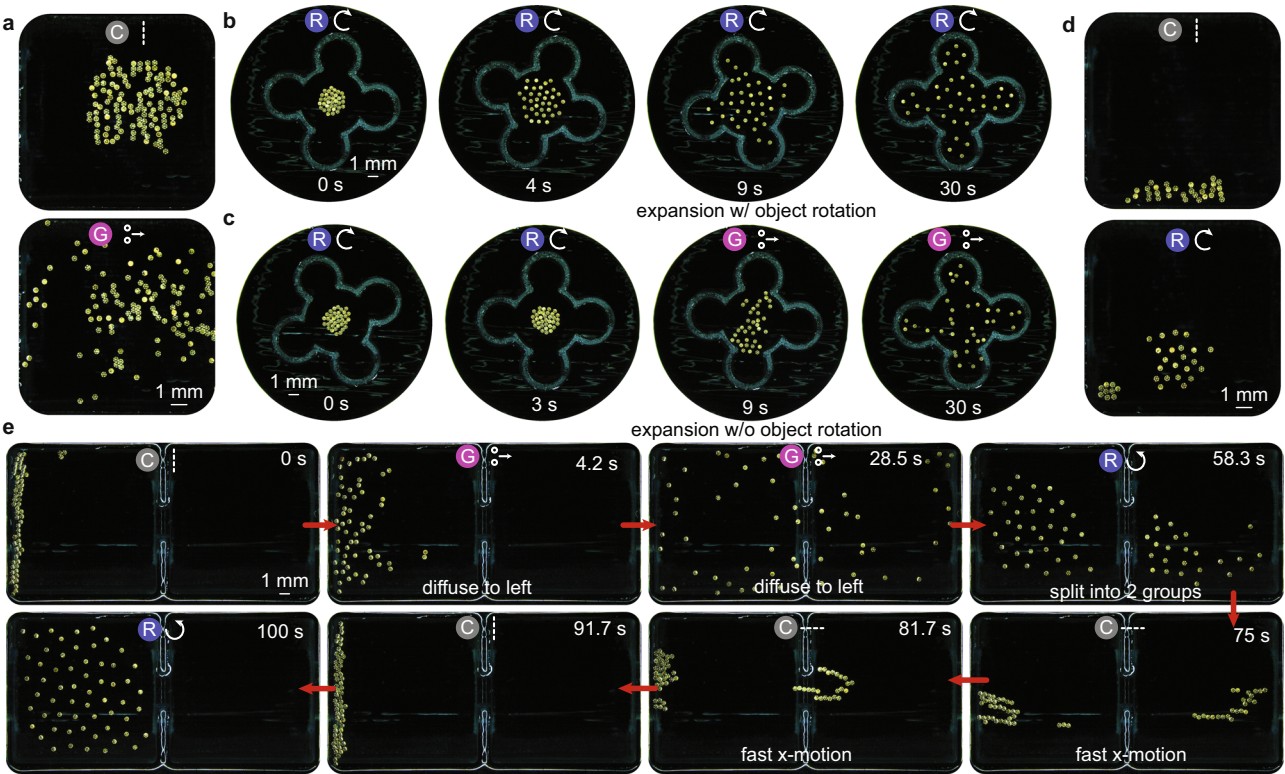

**Fig. 9 Dispersion and splitting of the microrobot collective. a** Representative images of a collective transitioning from Y chains to GaSPP to explore the available area. **b** Collective disperses using the rotation mode (70 Hz) within a quadrotor-shaped object; the collective expands and rotates the object about its center. **c** transition from rotation mode to GaSPP mode; the collective expands but does not rotate the object. **d** Collective splits into two groups by forming chains along a boundary and then transitioning to rotation mode. **e** Demonstration of a collective splitting between two sides through GaSPP and then returning to the same side through gradient-assisted X chain locomotion. The brightness of the images in (**a**–**e**) is enhanced using photoshop for better visualization. The symbols on top of the sub-images in (**a**–**e**) represent the mode the collective exhibits. These symbols correspond to those shown in Fig. 1a. The sub-images in (**b**, **c**) and (**e**) are labeled at the bottom with the function that the collective performs.

environments, contact-based and contact-free object transport, flow-induced object rotation, and collective dispersal to explore available areas and to split into smaller groups. These behaviors are possible due to the intricate balance between the mutual interactions between the micro-disks and the arena, and the judicious choice of the external magnetic field's form to drive the collective.

Although we explore two orthogonal oscillating magnetic fields in this study, we keep several parameters constant, like the phase difference between the two oscillating magnetic fields and the magnetic field intensity. Further variation of such parameters could uncover new behaviors in our system. One limitation of our system in its present state is its low force-bearing capacity, which limits the size of the object that can be pushed (via contact) using the chains. However, this problem could be solved either by increasing the collective size or through stronger magnetic field gradients. Moreover, the numerical model for GaSPP mode contains a noise term and its physical origin is fundamentally intriguing. Further studies on the physical origin of this noise is out of the scope of this study and will be reported separately.

Our system is versatile and demonstrates several useful behaviors and functions at this scale and at the fluid-air interface. In addition, two of the presented behaviors are unique to our system, i.e., the static collective composed of dynamic micro-disks and the gas-like formation composed of self-propelling pairs. Unlike any other system in the literature, our system transitions from globally driven to mutual interactions-dominated self-propelling behavior. Moreover, our system enables the use of traditional mechanical coupling at the

millimeter scale for torque transfer, paving a way for the transfer of knowledge of traditional mechanical systems to build machines at smaller length scales. One feature of our system is that even an incredibly small number of micro-disks, as small as 7, can exhibit all the formations demonstrated in this study. This capability could be very advantageous in scenarios when the collective needs to be broken into smaller groups while still retaining the ability to reconfigure. We envision that our system would be useful in performing several tasks applicable to microscale self-assembly and packaging[22], and its use as reconfigurable micromachines in biomedical and environmental applications.

## Methods

**Fabrication of the microrobots.** Microrobots were printed using two-photon polymerization-based 3D microprinting (Nanoscribe Photonic Professional GT) and nanofilms of 500 nm cobalt and 60 nm gold were then sputtered onto them using Kurt J. Lesker NANO 36.

**Video acquisition and analysis.** The experiments were performed under a Leica manual zoom microscope Z16 APO and they were recorded using a Basler acA2500-60uc camera. A LED light source SugarCUBE Ultra illuminator was connected to a ring light guide (0.83" ID, Edmund Optics #54-176) and was used to illuminate the micro-robots. A Python script was developed using the OpenCV library to process the experimental videos and extract the positions of the micro-robots. The analysis of the processed data was done using MATLAB. The raw images were used without any enhancements for the processing.

**Experimental protocol.** The characterization experiments (Fig. 4) were done in a square arena with a 12 mm side length. These experiments were perfomed using 120 micro-disks. For the rotation and oscillation mode, the experiments were

performed for $\Omega_{x,y}$ above the frequency range that results in aggregation ($\Omega_{x,y}$ >10 Hz) and below the step-out frequency ($\Omega_{x,y}$ ~65–75 Hz), where the external magnetic torque is insufficient for synchronized disk rotation. The experiments for static mode were performed for up to $\Omega_y = 60$ Hz with a step size of 5 Hz, since the disks start to step out at higher oscillation frequencies. At low frequencies ($\Omega_{x,y} < 10$ Hz), the capillary and magnetic dipole–dipole torques exceed the external magnetic field torque, while the attractive forces dominate over the repulsive hydrodynamic lift force, causing the micro-disks to align and connect along their corrugations, thereby forming aggregated clusters.

The experiments demonstrating the chain formations along any orientation ($\theta_{chain}$ with respect to the $x$ axis), the frequencies can be set to the linear relationship needed for a static collective ($\Omega_x = 2\Omega_y$) while $B_{x_1}$ and $B_{y_1}$ are varied according to the function $\theta_{chain} = \arctan(B_{y_1}/B_{x_1})$. For example, if $\theta_{chain}$ should be 45° below the $x$ axis, like the example shown in Supplementary Fig. 2b, then $\mathbf{B}$ could have the following parameters:

$$\mathbf{B} = \begin{bmatrix} 10\cos(40t) + 5 \\ 10\sin(20t) - 5 \end{bmatrix}. \tag{3}$$

The gear-like configuration (Fig. 8 f, g) was tested multiple times with the rings in physical contact and the result was the two rings rotating about their center of mass because of the increased capillary torque between the two rings.

**Simulations.** The numerical model developed by Wang et al.[35] was modified to simulate the different formations (see Methods subsection on model for simulations for more details on the numerical model). In the simulations, the initial positions of the micro-disks were chosen to be on a hexagonal lattice centered at the center of the frame. The initial distances between the neighboring micro-disks were 600 µm. The behaviors were simulated for a total time of 10 s with 1 ms steps. The equations described in the methods subsection on model for simulations were solved using the Explicit Runge-Kutta method of order 5(4) in the SciPy integration and ODEs library.

**Model for simulations.** The model used for simulations was adapted from literature[35] and modified to include the different magnetic field profile. The modified model is as follows:

If the center-center distance $r_{ji} >$ lubrication threshold ($= 315$ µm, or $1.1\ R$)

$$\frac{d\mathbf{r_i}}{dt} = \sum_{j\neq i} (6\pi\mu R)^{-1} \left( F_{mag-on,i,j}\left(r_{ji}, \phi_{ji}\right) + F_{cap,i,j}\left(r_{ji}, \phi_{ji}\right) + \frac{\rho\omega^2 R^7}{r_{ji}^3} \right) \cdot \hat{r}_{ji}$$
$$+ \sum_{j\neq i} \left( \frac{F_{mag-off,i,j}\left(r_{ji}, \phi_{ji}\right)}{6\pi\mu R} - \frac{R^3\omega}{r_{ji}^2} \right) \cdot \hat{r}_{ji} \times \hat{z} + \frac{\rho\omega_i^2 R^7}{6\pi\mu R} \cdot$$
$$\left( \left( \frac{1}{d_{toLeft}^3} - \frac{1}{d_{toRight}^3} \right) \cdot \hat{x} + \left( \frac{1}{d_{toBottom}^3} - \frac{1}{d_{toTop}^3} \right) \cdot \hat{y} \right), i = 1, 2, \ldots \tag{4}$$

$$\frac{d\alpha_i}{dt} = \frac{mB\sin(\theta-\alpha_i)}{8\pi\mu R^3} + \sum_{j\neq i} \frac{T_{mag-d,i,j}\left(r_{ji}, \phi_{ji}\right) + T_{cap,i,j}\left(r_{ji}, \phi_{ji}\right)}{8\pi\mu R^3}, i = 1, 2, \ldots \tag{5}$$

$$\mathbf{B}(t) = \left( B_{x_0}\cos\left(\Omega_x t\right) + B_{x_1} \right) \cdot \hat{x} + \left( B_{y_0}\cos\left(\Omega_y t\right) + B_{y_1} \right) \cdot \hat{y}, \tag{6}$$

where $\mathbf{r_i}$ and $\mathbf{r_j}$ are the position vectors of micro-disks;

$\mathbf{r_{ji}} = \mathbf{r_i} - \mathbf{r_j}$ is the vector pointing from the center of micro-disk $j$ to the center of micro-disks $i$;

$\alpha_i$ and $\alpha_j$ are the orientations of micro-disks;

$d$ is the edge-edge distance between two micro-disks;

$\phi_{ji}$ is the angle of dipole moment with respect to $\mathbf{r_{ji}}$. It is assumed to be the same for both micro-disks, as $\phi_{ji} = \phi_i = \phi_j$;

$\omega$ is the instantaneous spin speed of micro-disks;

$B = |\mathbf{B}|$ is the magnetic field strength (10 mT);

$\theta = \arctan(B_y/B_x)$ is the orientation of the external magnetic field;

$\Omega_x$ and $\Omega_y$ are the oscillation frequencies of the x and y component of the external magnetic field respectively;

$R$ is the radius of micro-disk (150 µm);

$\mu$ is the dynamic viscosity of water ($10^{-3}$ Pa·s);

$\rho$ is the density of water ($10^3$ kg/m³);

$m$ is the magnetic dipole moment of the micro-disks ($10^{-8}$ A·m²);

$F_{mag-on,i,j}$ and $F_{mag-off,i,j}$ are the magnetic dipole force on and off the center-to-center axis, respectively, and they are functions of $r_{ji}$ and $\phi_{ji}$;

$T_{mag-d,i,j}$ is the magnetic dipole torque, and it is a function of $r_{ji}$ and $\phi_{ji}$;

$F_{cap,i,j}$ is the capillary force, and it is a function of $r_{ji}$ and $\phi_{ji}$ and embeds the symmetry of a micro-disk;

$T_{cap,i,j}$ is the capillary torque, and it is a function of $r_{ji}$ and $\phi_{ji}$ and embeds the symmetry of a micro-disk;

$d_{toLeft}$, $d_{toRight}$, $d_{toBottom}$, and $d_{toTop}$ are the distances of a micro-disk to the four sides of the physical boundary.

If the center-center distance $r_{ji} <$ lubrication threshold ($= 315$ µm, or $1.1\ R$) and $r_{ji} \geq 300$ or $2R$,

$$\mu\frac{d\mathbf{r_i}}{dt} = \sum_{j\neq i} A\left(\frac{d_{ji}}{R}\right) \left( F_{mag-on,i,j}\left(r_{ji}, \varphi_{ji}\right) + F_{cap,i,j}\left(r_{ji}, \varphi_{ji}\right) + \frac{\rho\omega^2 R^7}{r_{ji}^3} \right)\hat{r}_{ji}$$
$$+ \sum_{j\neq i} B\left(\frac{d_{ji}}{R}\right) F_{mag-off,i,j}\left(r_{ji}, \varphi_{ji}\right)\hat{r}_{ji}\times\hat{z}$$
$$+ \sum_{j\neq i} C\left(\frac{d_{ji}}{R}\right) mB\sin(\theta-\alpha_i)\hat{r}_{ji}\times\hat{z} + \frac{\rho\omega_i^2 R^7}{6\pi R}\left( \left( \frac{1}{d_{toLeft}^3} - \frac{1}{d_{toRight}^3} \right)\hat{x} \right.$$
$$+ \left.\left( \frac{1}{d_{toBottom}^3} - \frac{1}{d_{toTop}^3} \right)\hat{y} \right), i = 1, 2, \ldots \tag{7}$$

$$\mu\frac{d\alpha_i}{dt} = G\left(\frac{d_{smallest}}{R}\right) mB\sin(\theta-\alpha_i) + \sum_{j\neq i} G\left(\frac{d_{ji}}{R}\right) T_{mag-d,i,j}\left(r_{ji}, \varphi_{ji}\right) + T_{cap,i,j}\left(r_{ji}, \varphi_{ji}\right), i = 1, 2, \ldots \tag{8}$$

$$\mathbf{B}(t) = \left( B_{x_0}\cos\left(\Omega_x t\right) + B_{x_1} \right) \cdot \hat{x} + \left( B_{y_0}\cos\left(\Omega_y t\right) + B_{y_1} \right) \cdot \hat{y}, \tag{9}$$

where the coefficients $A(x)$, $B(x)$, $C(x)$ and $G(x)$ are lubrication coefficients.

If the center-center distance $r_{ji} < 300$ or $2R$, a repulsion term is added to the force equation,

$$\mu\frac{d\mathbf{r_i}}{dt} = \sum_{j\neq i} A(\varepsilon)\left( F_{mag-on,i,j}\left(2R, \varphi_{ji}\right) + F_{cap,i,j}\left(2R, \varphi_{ji}\right) + \frac{\rho\omega^2 R^7}{r_{ji}^3} \right)\hat{r}_{ji} + \sum_{j\neq i}\frac{F_{wallRepulsion}}{6\pi R}\cdot\frac{-d_{ji}}{R}\hat{r}_{ji}$$
$$+ \sum_{j\neq i} B(\varepsilon)F_{mag-off,i,j}\left(2R, \varphi_{ji}\right)\hat{r}_{ji}\times\hat{z} + \sum_{j\neq i} C(\varepsilon)mB\sin(\theta-\alpha_i)\hat{r}_{ji}\times\hat{z}$$
$$+ \frac{\rho\omega_i^2 R^7}{6\pi R}\left( \left( \frac{1}{d_{toLeft}^3} - \frac{1}{d_{toRight}^3} \right)\hat{x} + \left( \frac{1}{d_{toBottom}^3} - \frac{1}{d_{toTop}^3} \right)\hat{y} \right), i = 1, 2, \ldots \tag{10}$$

$$\mu\frac{d\alpha_i}{dt} = G(\varepsilon)mB\sin(\theta-\alpha_i) + \sum_{j\neq i} G(\varepsilon)T_{mag-d,i,j}\left(2R, \varphi_{ji}\right) + T_{cap,i,j}\left(2R, \varphi_{ji}\right), i = 1, 2, \ldots \tag{11}$$

$$\mathbf{B}(t) = \left( B_{x_0}\cos\left(\Omega_x t\right) + B_{x_1} \right) \cdot \hat{x} + \left( B_{y_0}\cos\left(\Omega_y t\right) + B_{y_1} \right) \cdot \hat{y}, \tag{12}$$

where $\varepsilon$ is a small number ($10^{-10}$ µm/R); $F_{wallRepulsion}$ is set to be $10^{-7}$ N.

**Model for GaSPP simulations.** To simulate the Gas-like mode containing self-propelling pairs (GaSPP), following modifications were made to the model in the section on model for simulations:

$$\frac{d\mathbf{r_i}}{dt} = \sum_{j\neq i} (6\pi\mu R)^{-1}\left( F_{mag-on,i,j}\left(r_{ji}, \phi_{ji}\right) + F_{cap,i,j}\left(r_{ji}, \phi_{ji}\right) + \frac{\rho\omega^2 R^7}{r_{ji}^3} \right) \cdot \hat{r}_{ji}$$
$$+ \sum_{j\neq i}\left( \frac{F_{mag-off,i,j}\left(r_{ji}, \phi_{ji}\right)}{6\pi\mu R} - \frac{R^3\omega}{r_{ji}^2} \right) \cdot \hat{r}_{ji}\times\hat{z} + \frac{\rho\omega_i^2 R^7}{6\pi\mu R}\cdot$$
$$\left( \left( \frac{1}{d_{toLeft}^3} - \frac{1}{d_{toRight}^3} \right) \cdot \hat{x} + \left( \frac{1}{d_{toBottom}^3} - \frac{1}{d_{toTop}^3} \right) \cdot \hat{y} \right), i = 1, 2, \ldots \tag{13}$$

$$\frac{d\alpha_i}{dt} = \frac{mB\sin(\theta-\alpha_i)}{8\pi\mu R^3} + \sum_{j\neq i}\frac{T_{mag-d,i,j}\left(r_{ji}, \phi_{ji}\right) + T_{cap,i,j}\left(r_{ji}, \phi_{ji}\right)}{8\pi\mu R^3}, i = 1, 2, \ldots \tag{14}$$

$$\alpha_i(t) = \int \frac{d\alpha_i}{dt} \cdot d\tau + \eta_i(t) \tag{15}$$

$$\mathbf{B}(t) = \left( B_{x_0}\cos\left(\Omega_x t\right) + B_{x_1} \right) \cdot \hat{x} + \left( B_{y_0}\cos\left(\Omega_y t\right) + B_{y_1} \right) \cdot \hat{y}, \tag{16}$$

where $\eta_i$ is a gaussian white noise with mean 0 and standard deviation 10°.

The simulations for GaSPP mode does not agree with the experiments perfectly. While the simulations reproduce the breaking up of the collective into self-propelling pairs, the ballistic motion of the pairs is shorter in duration in the simulations as compared to the experiments.

**Calculation of hexatic order parameter.** The local hexatic order parameter was calculated as:

$$\psi_{6_{Local}} = \frac{\Sigma_k\left|e^{i6\theta_k}\right|}{K}, \tag{17}$$

where $K$ is the total number of neighbors of all micro-disks; $k$ is neighbor index, $\theta_k$ is the polar angle of the vector from each micro-disk to the $k^{th}$ neighbor. The global

hexatic order parameter was calculated as:

$$\psi_{6_{Global}} = \left| \frac{\Sigma_k e^{i6\theta_k}}{K} \right|. \tag{18}$$

**Dye for flow visualization experiments.** Fluorescein sodium ($C_{20}H_{10}Na_2O_5$) dye from Fischer scientific (code: F/1300/48) was used to visualize the azimuthal flow around the rotating collective.

**Reporting Summary.** Further information on research design is available in the Nature Research Reporting Summary linked to this article.

## Data availability

The data needed to evaluate the conclusions of this study are present in the paper and its supplementary information files. All data are available from the corresponding author upon reasonable request.

## Code availability

The codes are available in open source in Zenodo[60].

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

## Acknowledgements

We thank G. Ricther and F. Thiele for help on sputtering, T. Wang for 3D printing of arenas, N. K. Subbaiah for help on Nanoscribe 3D micro-printing, R. Soon for discussions on bode plot calculations and C. Holm for discussions on numerical model for the simulations. Funding: This work is funded by the Max Planck Society. G.G. thanks the International Max Planck Research School for Intelligent Systems for financial support. S.C and K.P thank the National Science Foundation Graduate Research Fellowship, the Fulbright Germany Scholarship, the National Science Foundation grant 2042411, the Packard Foundation Fellowship for Science and Engineering, and the Aref and Manon Lahham Faculty Fellowship. W.W. thanks UM-SJTU start-up fund and National Science Foundation of China general program 22175115.

## Author contributions

G.G. and S.C. designed, performed, and analyzed the experiments; W.W. and G.G. contributed to the construction of the pairwise numerical models; G.G. fabricated the micro-disks; G.G., S.C. and W.W. contributed to data processing and analysis; G.G. and S.C. wrote the manuscript; W.W. contributed to the design of the experimental setup; all authors contributed to the conception and contextualization of the work; all authors discussed the results and contributed to the editing of the manuscript; K.P. and M.S. supervised the research.

## Funding

## Competing interests

The authors declare no competing interests.
