## [Peer Review File · Nature Communications]

REVIEWER COMMENTS

Reviewer #1 (Remarks to the Author):

This paper presents a series of collective behaviours using magnetic micro-disks (120) at the air-water interface controlled by two independent oscillating magnetic fields. The six behaviours include rotation, oscillation, static, chains, oscillating chains, and a new gas-like mode which shows self-propelling pairs. Behaviours are driven by the global magnetic fields, but also particle-particle and particle-fluid interactions (e.g., hydrodynamic, capillary, and magnetic dipole-dipole interactions). They show they can dynamically change between these behaviours, and that they can be combined to perform useful functions including collective navigation in complex environments, precise motion control (to spell out letters for example), and transport of spheres or scaffolds. Two of the behaviours are novel, the static mode and gas-like mode.

Overall this is an impressive collection of behaviours and combinations of behaviours to demonstrate function. The key novelty lies in the breadth and versatility of the behaviours demonstrated. The paper is well written, with mostly clear figures and helpful videos to show the dynamics of the system. The effort to back experiments with simulations is also appreciated.

The following points should be considered to improve the manuscript.

The two new behaviours, static mode and GaSPP are interesting, especially GaSPP which transforms particle pairs into active self-propelling systems. The value of these two behaviours is not fully demonstrated through the functionalities shown in Fig 5-8. In Figure 8, GaSPP is not quite helpful other than for exploration, which could perhaps be done through repulsion. It is not widely used in the other demonstrations. I believe there is great potential for this behaviour, perhaps this could be further motivated in the text by saying how GaSPP could be used?

In general, many behaviours are shown to achieve Fig 5-8. Could the same functions (e.g. navigation, transport) be achieved with fewer modes of operation? Sometimes it feels like modes are being used for the sake of it. This does have demonstration value, but perhaps better motivating the need for the multiple functions would be useful. Could contact-based transport be achieved with just Y chains (rather than X and Y). Could contact-free transport be achieved with rotation alone? One improvement could be to label the sub-images with what mode is being used and for what function (e.g. Mode: Chain - Function: Collective motion - or a more compact version of this). Fig 1 is also quite loaded and would deserve additional labels to describe what functions/behaviours are taken at each step (e.g. transport, collective motion, manipulation, dispersion, etc). Perhaps further segmentation/labelling 1b could help?

The overall introduction is a bit too loose, with discussions of reconfigurability, self-assembly, formation, self-organisation, and collective behaviour. The abstract mentions swarming 3 times - swarms are then never mentioned in the text (collective behaviour fits better given the global control). There are examples from nature (slime mode, bacteria), robotics, and soft matter. This makes it slightly difficult to grasp the scope of the paper early on. Perhaps look at narrowing down the introduction and making it more precise with similar terminology throughout.

The following points are minor:

The paper mentions "macroscale robot collectives that exploit both physical and explicit interactions among agents" - It's not clear what this means. Physical interactions can be explicit - or perhaps physical here means environmental?

"At this scale, systems function through physical and chemical interactions to create formations that can manipulate larger objects". - It's not clear why the focus at this point in the introduction is manipulation.

"The key difference here is that the micro-disks' collective behaviours are determined by controllable global stimuli, whereas active Janus particles are mainly driven by local stimuli." - does this still qualify as an active particle given that it's controlled globally. In general, a bit more information on how the GaSPP works would be help, especially given this is one of the key novelties of the paper.

In Fig4, the simulations don't seem to capture very well the final conditions for Rotation ($\Omega_x, \Omega_y=70$ Hz), Static ($\Omega_x=60$ Hz, $\Omega_y=30$ Hz). This is perhaps worth discussing.

Reviewer #2 (Remarks to the Author):

In this paper, the authors presented a microrobot collective system that can reconfigure into six different behaviors, each of which is capable of diverse tasks. And, they showed the different behaviors our system exhibits, both via experiments and simulations. And, they also characterized each behavior at different frequencies and demonstrate inter-mode transitions, and realized several functions like collective navigation through confined environments. The contents of the paper are very interesting and well described. However, the following issues should be considered and addressed in the manuscript.

1. The raft (micro-disk in the paper) has a round, hexatic capillary shape. Are there some reasons for that specific shape?

2. In the chain modes, the magnetic field oscillates by time. The magnetic particles can form chains with the static magnetic field. The addition of some comments for the reason of applying an oscillating magnetic field instead of the static magnetic field would be good.

3. In the case of the high speed of the raft itself, the rafts collide with each other. Is there some consideration of physical repulsion in the simulation model needed for avoiding any overlaps between rafts? If so, the notion of the consideration would be good.

4. In the chain motions of Fig. 4, there are some differences between the experiments and simulations. The reason for the differences should be discussed.

5. In Fig. 5 a (ii), the magnetic field gradient's unit is G/mm². Please double-check the unit of the field gradient.

6. The coil system set up in the paper generates some tens of mT magnetic fields in about a hundred frequencies. Some confirmation of the capability of the generation of dynamic magnetic fields of the setup e.g. bode plot of current in the coil system would improve the paper.

Reviewer #3 (Remarks to the Author):

This manuscript entitled 'Microrobot Collectives with Reconfigurable Morphologies, Behaviors, and Functions' presents a collective microrobot system that shows different behaviors under actuation of externally magnetic field. Meanwhile, the inter-mode transitions and various tunability of the swarm are also demonstrated by experiments and simulations. As we all know, the microrobot swarms show the advantage of faster speed, higher carrying efficiency and group deformation, which could overcome the disadvantage of low delivery efficiency and slow speed of a single microrobot. Moreover, microrobot swarm should be composed of multiple microrobots with motion behavior. However, the motion behavior of a single micro-disk under different magnetic fields was not exhibited. Then, the control modes and six motion forms under different magnetic fields in the manuscript do not show any new ideas · highly similar with the theme of the previous publications (Nature Communication (2019) 10:5631; Sci. Robot. 4, eaav8006 (2019)). Above all, in terms of application, transporting and rotating objects shown by the microrobot swarm are not bright spots. Hence, I cannot recommend the publication of this manuscript.

Response Letter

We thank the reviewers for their helpful and constructive comments on our manuscript. Below is our response to each reviewer's comments. The original comments are in italic, and our response are indented and in normal text in dark blue color. The new data and edits in the revised manuscript and SI are highlighted in yellow.

Reviewer #1:

This paper presents a series of collective behaviours using magnetic micro-disks (120) at the air-water interface controlled by two independent oscillating magnetic fields. The six behaviours include rotation, oscillation, static, chains, oscillating chains, and a new gas-like mode which shows self-propelling pairs. Behaviours are driven by the global magnetic fields, but also particle-particle and particle-fluid interactions (e.g., hydrodynamic, capillary, and magnetic dipole-dipole interactions). They show they can dynamically change between these behaviours, and that they can be combined to perform useful functions including collective navigation in complex environments, precise motion control (to spell out letters for example), and transport of spheres or scaffolds. Two of the behaviours are novel, the static mode and gas-like mode.

Overall this is an impressive collection of behaviours and combinations of behaviours to demonstrate function. The key novelty lies in the breadth and versatility of the behaviours demonstrated. The paper is well written, with mostly clear figures and helpful videos to show the dynamics of the system. The effort to back experiments with simulations is also appreciated.

Response: We thank the reviewer for highlighting the two novel behaviors of our system and for recognising the potential for the GaSPP behavior.

The following points should be considered to improve the manuscript.

1. The two new behaviours, static mode and GaSPP are interesting, especially GaSPP which transforms particle pairs into active self-propelling systems. The value of these two behaviours is not fully demonstrated through the functionalities shown in Fig 5-8. In Figure 8, GaSPP is not quite helpful other than for exploration, which could perhaps be done though repulsion. It is not widely used in the other demonstrations. I believe there is great potential for this behaviour, perhaps this could be further motivated in the text by saying how GaSPP could be used?

Response: We thank the reviewer for highlighting the two novel behaviors of our system and for recognising the potential for the GaSPP behavior.

The following text has been added to the section on GaSPP mode:

The GaSPP mode opens up possibilities for several fundamental studies investigating the systems of active particles. For example, studies like those investigating the emergence of directed motions in a collective of active particles⁵⁶, could also be possible using the GaSPP mode, by tuning the density of the micro-disks in GaSPP mode (the speed of the pairs can be

tuned using the external magnetic field) to study the emergence phenomenon in such systems. The GaSPP mode could be advantageous over other systems, like the quince rollers, or the active janus particles, because the mutual interactions among the micro-disks in GaSPP mode are relatively simpler than the non-trivial fluidic interactions among the quince rollers and also GaSPP does not require any fuel (as is the case for the chemotactic janus particles). Moreover, the speed of the pairs in the GaSPP can be tuned by the external magnetic field. The GaSPP behavior could also be useful for demonstrating theories such as the long-range order in active systems. The GaSPP mode can be useful for robotics studies as well. For example, the GaSPP can be used as the microrobot swarm to create superstructures like the ones developed using the rodlike vibrating robots^{57,58}.

The following is the modified beginning of the ‘Dispersion and splitting’ subsection in the ‘Mode-Enabled Robotic Functionalities’ section with the following two sentences:

More specifically, this mode enables small collectives to disperse across large areas, within an object without transferring a significant amount of torque, as well as escape a free-floating structure without changing its orientation.

The following lines were added at the end of the ‘Dispersion and splitting’ subsection to highlight how GaSPP could be used for fundamental studies on ‘self-propelling’ collectives:

Aside from an application perspective, this novel mode has great potential in carrying out fundamental studies on self-propelling particles; although the behavior is enabled by mutual interactions between particle pairs, their speed is tuneable through a global magnetic field frequency. Studying the fundamental behaviors of this mode more closely could give insight into the possibilities of self-propelling collectives like those composed of Janus particles.

2. In general, many behaviours are shown to achieve Fig 5-8. Could the same functions (e.g. navigation, transport) be achieved with fewer modes of operation? Sometimes it feels like modes are being used for the sake of it. This does have demonstration value, but perhaps better motivating the need for the multiple functions would be useful. Could contact-based transport be achieved with just Y chains (rather than X and Y). Could contact-free transport be achieved with rotation alone? One improvement could be to label the sub-images with what mode is being used and for what function(e.g. Mode: Chain - Function: Collective motion - or a more compact version of this). Fig 1 is also quite loaded and would deserve additional labels to describe what functions/behaviours are taken at each step (e.g. transport, collective motion, manipulation, dispersion, etc). Perhaps further segmentation/labelling 1b could help?

Response: We thank the reviewer for this suggestion. Indeed, in some demonstrations, the use of certain modes could have been avoided but they were kept for the convenience of performing the experiments and their demonstration value. For example, the first two steps were the same for Fig. 5e and Fig. 6c - switching from static to x-chains to reach the bead in the right, and then either y-chains or rotation mode were used for contact-based or contact-free transport, respectively. The first two steps were kept similar to show that the bead can be reached using the same sequence of modes, while the method of transportation depends on whether the y-chains or the rotation mode is used. However, we also used multiple behaviors to highlight a complete

picture in most demonstrations. We explain why different modes were used in various demonstrations in the following:

In Figs. 5e and 6c, X-chains were used to move the collective from the left side to the right side of the arena; locomotion under an external magnetic field gradient is faster in the x-direction when the collective is in X-chains than any other mode. We could use another mode to locomote but X-chains maximize the collective's speed and also minimize the contact area of the collective on the bead as the collective is passing it. Also, we could have started the collective near the bead and switched to Y-chains to transport the bead using contact-based pushing or switched to rotation mode for contact-free transport. However, the collective may not always be near the bead that needs to be transported. Therefore, to present a complete picture, we designed a demonstration where the collective started farther from the bead, as such a demonstration could highlight the advantage of using multiple modes even to achieve a single function. Also, in Fig. 6c, the static mode was used because it produces minimal or no flow around it and that is beneficial in stopping the collective without a collateral effect on the bead.

In Fig. 7h, the static mode was used to gather the disks inside the C-shaped object. The static mode was used because it is the most cohesive (as evident from Fig. 3c) at any spin speed and because it produces negligible flow around itself for $\gamma > 30\text{Hz}$ (as evident from Fig. 3b). The GaSPP mode was used in the end to escape the object because unlike the rotation mode at 70 rps, the GaSPP mode enables the disks to escape the object without changing the orientation of the object.

In Fig. 8d, x-chains were used to move the collective from right to left because the collective moves the fastest in x-direction when in x-mode.

The following text has been added to the 'Gradient-based locomotion and navigation through intricate environments' subsection to motivate the use of the rotation mode, X-chain, and Y-chain modes throughout different regions of the arena:

Additionally, the collective moves quickest as a chain, when the chain is aligned along the direction of the magnetic field gradient.

The following text has been added to the 'Contact-based object transport' subsection in the 'Mode-Enabled Robotic Functionalities' section to motivate the switching between X- and Y-chains to transport the object:

It is important that the collective exhibits the X-chain mode when moving to the right side of the arena since this allows it to maximize its speed and minimize the contact surface area to prevent pushing the bead to the right side.

The following text has been added to the 'Contact-free object transport' subsection in the 'Mode-Enabled Robotic Functionalities' section to motivate the use of X-chains and static mode before transporting the object through rotation mode:

Similarly to the experiments in Fig. 5f, the collective navigates to the right side of the arena and around the bead. As opposed to the demonstration in Fig. 5f where the collective must be aligned

adjacent to the bead to transport it, here the collective must encapsulate the bead so there is a significant density of micro-disks around it in all directions. After locomoting to the arena's right side at a high speed through X-chains, the collective switches to static mode so that micro-disks can evenly distribute throughout the bead's perimeter without transferring torque that might send the bead in an undesired direction.

We have labeled the sub-figures in Fig. 1 and Figs. 5-8 according to the reviewer's suggestion. The updated figures are added below:

Fig. 1. Microrobot collective reconfigurable behaviors and functions. (a) Three micro-disks driven by external magnetic fields exhibit magnetic dipole-dipole attraction, capillary attraction, and hydrodynamic repulsion (top). Six collective modes are possible: rotation (R), oscillation (O), static (S), chains (C), oscillating chains (OC), and gas-like mode containing self-propelling pairs (G) (bottom). (b) Schematic of collective transitioning between the possible modes to perform various functions. Starting on the left side and continuing clockwise along the edge: the collective starts out in static mode and transitions to a chain to locomote through a narrow channel and exit the channel to form oscillating chains (channel crossing). At the top, the collective lines itself up against a wall and uses the physical interactions with the wall to separate into two clusters when it transitions to rotation mode (splitting). The collective can then pass around an obstacle more easily by adapting inter-disk distance through its static mode at high magnetic field frequencies (adapting to environment). The collective can then rotate and locomote at the same time (generating azimuthal flow) and induce motion on surrounding objects through its azimuthal flow field (object rotation and flow-based transport). At the bottom, the collective forms chains and pushes on an object (contact-based transport) and then disperses through a gas-like mode (dispersion and exploration). The center images show the collective can rotate objects through the azimuthal flow field when the micro-disks are within the object as well as encapsulating it.

Fig. 5. Magnetic gradient-assisted collective navigation through confined environments and contact-based object transport. (a) Locomotion speed comparison for the rotation, static, oscillation, and chain modes with $N = 126$ micro-disks driven at different frequencies when the gradient is $F = 0.7$ Gauss / mm. The chains move faster than other modes. The error bars represent standard deviations over 5 s. (b) Displacement of each mode over time when 30 Hz. The chains move faster and get slowed down due to the physical boundary around 2.5 s. This contributes to large standard deviations in chain speed in (a). (c) Representative images showing the trajectories of static (left), oscillating (middle left), rotating (middle right), and chain (right) collectives under the influence of a magnetic field gradient along the y axis. (d) Y chain of seven micro-disks driven with magnetic field gradients to produce MPI and C trajectories (Supplementary Movie 7). (e) 17 micro-disks transition between different modes to navigate through narrow passages (Supplementary Movie 8). (f) The collective switches between the modes static, X chains, and Y chains to locomote and push an object. The brightness of the images in (c-f) is enhanced using photoshop for better visualization. The symbols on the sub-images at the bottom, in (c), and top, in (e-f), represent the mode the collective exhibits. These symbols correspond to those shown in Fig. 1a. The sub-images in (f) are labeled at the bottom with the function that the collective performs.

Fig. 6. Flow-assisted contact-free object transport. (a) Qualitative demonstration of flow around micro-disks when the collective is rotating. Green and blue lines were extracted using edge detection (see Supplementary Movie 10 for the unprocessed video). (b) Collective rotates to guide a purple ball to the bottom right-hand corner. (c) Collective rotation guides a ball around the perimeter of the arena by using the arena design. The brightness of the images in (a-c) is enhanced using photoshop for better visualization. The symbols on top of the sub-images in (b-c) represent the mode the collective exhibits. These symbols correspond to those shown in Fig. 1a. The sub-images in (b-c) are labeled at the bottom with the function that the collective performs.

Fig. 7. Flow-assisted object rotation and orientation control. (a) Micro-disk collectives within ring-shaped objects of different sizes; the object rotates in the same direction as the collective. (b) Collectives encapsulate and rotate ring-shaped objects; the object rotates in the opposite direction as the collective. Red line in (a-b) shows the rotational trajectory of the rings. The same rings are used for (a) and (b). (c) Angular velocity of a ring-like object (ID 3.4 mm and OD 6.5 mm) when there are 60 micro-disks inside or outside the structure. Error bars represent standard deviations over 10 s. (d) Collective rotates a rod-shaped object and (e) a star-shaped object. (f) Two collectives within adjacent objects enable coupled object rotation. (g) A collective within one of two adjacent ring-like objects rotates the structure it is in and exerts a torque on the neighboring object to enable gear-like motion. (h) A collective enters a C-shaped object, rotates it, and exits the structure. A red circle is added to highlight the orientation of the rings (a-b) and (g-f). The red dashed arrows in (d-g) show the direction of rotation of the individual objects (thin arrows) and their center of mass (thick arrows). The brightness of the images in (a-b) and (d-h) is

enhanced using photoshop for better visualization. The symbols on top of the sub-images in (a-b) and (h) represent the mode that the collective exhibits. These symbols correspond to those shown in Fig. 1a. The sub-images in (h) are labeled at the bottom with the function that the collective performs.

Fig. 8. Dispersion and splitting of the microrobot collective. (a) Representative images of a collective transitioning from Y chains to GaSPP to explore the available area. (b) Collective disperses using the rotation mode (70 Hz) within a quadrotor-shaped object; the collective expands and rotates the object about its center. (c) transition from rotation mode to GaSPP mode; the collective expands but does not rotate the object. (d) Collective splits into two groups by forming chains along a boundary and then transitioning to rotation mode. (e) Demonstration of a collective splitting between two sides through GaSPP and then returning to the same side through gradient-assisted X chain locomotion. The brightness of the images in (a-e) is enhanced using photoshop for better visualization. The symbols on top of the sub-images in (a-e) represent the mode the collective exhibits. These symbols correspond to those shown in Fig. 1a. The sub-images in (b-c) and (e) are labeled at the bottom with the function that the collective performs.

3. The overall introduction is a bit too loose, with discussions of reconfigurability, self-assembly, formation, self-organisation, and collective behaviour. The abstract mentions swarming 3 times - swarms are then never mentioned in the text (collective behaviour fits better given the global control). There are examples from nature (slime mode, bacteria), robotics, and soft matter. This makes it slightly difficult to grasp the scope of the paper early on. Perhaps look at narrowing down the introduction and making it more precise with similar terminology throughout.

Response: We thank the reviewer for the suggestion. We have corrected the inconsistencies in the text and updated the introduction.

First, we have taken out the word swarm in the abstract and replaced it with the word ‘collective’ which we use throughout the remainder of the text and represents our system more accurately. The modified lines from the abstract are added below:

Many self-organizing microrobotic collectives have been developed to overcome inherent limits in actuation, sensing, and manipulation of individual microrobots; however, reconfigurable collectives with robust transitions between behaviors are rare. Such systems that perform multiple functions are advantageous to operate in complex environments.

Second, we have updated the introduction by narrowing down our description of biological collectives and focusing on macro / micro robot collectives relevant to our study. The following are the updated lines in the opening paragraph of the introduction:

Collectives in nature often make use of reconfiguration, altering the group's morphology to carry out complex functions in various environments¹⁻⁷. At small scales, reconfiguration enables organisms to adapt to environmental disturbances and complete diverse functions. Inspired by the robustness and adaptability of these systems, engineers have mimicked naturally occurring behaviors through robot collectives that are programmable and interact with their environment to enable robust reconfiguration. For example, at the macro-scale, Kilobot collectives use programmed interactions to create different formations⁸⁻¹⁰ and reconfigure to manipulate objects based on global signal inputs¹¹. Other macro-scale robot collectives demonstrate how environmental interactions like contact-based coupling¹²⁻¹⁵ can enable collectives to change their shape¹⁶⁻¹⁸, function, and mode of locomotion¹⁹⁻²¹. At the micron-scale, reconfiguration in artificial systems is rare and it has the potential to open up possibilities in biomedicine, environmental remediation, and other applications^{22,23}. At this scale, collectives interact through physical and chemical interactions to organize global responses beyond the reach of individuals²⁴⁻²⁸.

The following lines are added at the end of the first paragraph of the introduction:

Therefore, a system that can utilize the mutual interactions between its constituents and respond to a global magnetic field stimuli to exhibit different behaviors would help realize a versatile reconfigurable robot collective.

The following points are minor:

4. *The paper mentions "macroscale robot collectives that exploit both physical and explicit interactions among agents" - It's not clear what this means. Physical interactions can be explicit - or perhaps physical here means environmental?*

Response: We thank the reviewer for highlighting the confusion. Our previous sentence did take ‘physical’ to mean ‘environmental’, and we believe changing ‘explicit’ to ‘programmed’ will make it easier for readers to understand that we are referring to robots that sense their

surroundings, process the information, and execute an appropriate behavior to the current situation. We have updated the manuscript text so it reads:

robot collectives that are programmable and interact with their environment to enable robust reconfiguration.

5. *"At this scale, systems function through physical and chemical interactions to create formations that can manipulate larger objects". - It's not clear why the focus at this point in the introduction is manipulation.*

Response: We have corrected the line as follows:

At this scale, collectives interact through physical and chemical interactions to organize global responses beyond the reach of individuals.

6. *"The key difference here is that the micro-disks' collective behaviours are determined by controllable global stimuli, whereas active Janus particles are mainly driven by local stimuli." - does this still qualify as an active particle given that it's controlled globally. In general, a bit more information on how the GaSPP works would be help, especially given this is one of the key novelties of the paper.*

Response: The reviewer is right in pointing out that the disks in GaSPP are controlled globally and therefore it raises a question to the disks in GaSPP being considered active. Our reasoning for considering them active is the following:

The disks only translate when they are in pairs. An unpaired disk does not translate at all. Therefore the mutual interactions between the disks must play a significant role in the translation of the disks in GaSPP. The frequency of the global magnetic field affects the translation speed of the pairs. However, the direction of translation of different pairs is random and seems to change randomly. The pairs behave like active brownian particles (stochastic trajectories with long persistence length). Therefore, noting that there is an influence of the local environment around the disks in GaSPP (paired disks translate in random directions and an unpaired disk, with neighbors far away, does not translate) the disks may be considered active.

The following lines are added to the section on GaSPP to explain the mechanism for GaSPP formation:

When two disks spinning in opposite directions come close, the net azimuthal flow created by them causes them to translate as a pair, just in the same way as when two disks spinning in the same direction (in the rotation mode), start orbiting around their common center of mass. When the external 1D oscillating magnetic field that enables GaSPP, changes its direction, it makes an angle of 180° with the magnetic dipole on the disks, creating an unstable equilibrium. Because of this unstable equilibrium, the micro-disks can either spin clockwise or counter-clockwise with equal probability, to align their magnetic dipole with the external magnetic field. Thus, the collective breaks into pairs composed of one disk each that spins in either direction. These pairs translate in random directions and their translation speed can be controlled by the frequency of the oscillating magnetic field.

7. In Fig4, the simulations don't seem to capture very well the final conditions for Rotation ($\Omega_x, \Omega_y=70$ Hz), Static ($\Omega_x=60$ Hz, $\Omega_y=30$ Hz). This is perhaps worth discussing.

Response: We thank the reviewer for the suggestion. This disagreement is because of the following reason:

We consider the drag from the physical boundary to be insignificant compared to the other interactions in the system and therefore we do not model it in our numerical simulations. However, this drag becomes significant for higher angular speed of the microdisks (at higher oscillation frequencies of external magnetic fields for rotation and static modes). Due to the lack of the drag term from the physical boundary in our model, the numerical simulations cannot reproduce the exact final conditions for the rotation and static mode at higher oscillation frequencies. We have added the following lines to the main text to discuss the disagreement pointed out by the reviewer:

Note that there are some discrepancies between experiments and simulations in the final formations for the rotation mode ($x, y=70$ Hz) and static mode ($x=60$ Hz, $y=70$ Hz). These discrepancies arise because we do not consider the hydrodynamic drag from the arena boundary, which remains insignificant in most cases but becomes significant at higher frequencies of the external magnetic field. Because the modeling of the drag from the arena boundary is highly non-trivial and because the absence of this term does not influence the simulation results significantly, this term was not included in our numerical model.

Reviewer #2:

In this paper, the authors presented a microrobot collective system that can reconfigure into six different behaviors, each of which is capable of diverse tasks. And, they showed the different behaviors our system exhibits, both via experiments and simulations. And, they also characterized each behavior at different frequencies and demonstrate inter-mode transitions, and realized several functions like collective navigation through confined environments. The contents of the paper are very interesting and well described. However, the following issues should be considered and addressed in the manuscript.

Response: We thank the reviewer for the positive assessment of our work. We answer the issues raised by the reviewer individually in what follows.

1. The raft (micro-disk in the paper) has a round, hexatic capillary shape. Are there some reasons for that specific shape??

Response: The six symmetrically placed cosine profiles along the edge of the micro-disks create capillary interactions (with six-fold symmetry) between any two micro-disks in the system. The behaviors in the system would not be affected significantly if micro-disks with a different symmetry (between two and six) or a different shape were used instead. However, modeling the mutual interactions between the disks with a complex shape involves unnecessary complications that could be avoided by using a circular shape with six-fold symmetry, thereby allowing us to explore the characteristics and functions of each of the reconfigurable collective behaviors in-depth. We would like to highlight that it is the intricate balance among the different mutual interactions between the micro-disks that enable different behaviors of the system. Eliminating any of the mutual interactions will result in a less versatile system.

The following text was added in the opening paragraph of the ‘Modes of Collective Behavior’ section to address this comment:

The micro-disks’ hexatic characteristic causes capillary interactions with six-fold symmetry. The number of symmetrically-placed sinusoidal profiles (between 2-6) would not significantly affect the behaviors presented in this work, however, the six-fold symmetry and circular shape are used because of simplicity in modeling the mutual interactions between such disks.

2. In the chain modes, the magnetic field oscillates by time. The magnetic particles can form chains with the static magnetic field. The addition of some comments for the reason of applying an oscillating magnetic field instead of the static magnetic field would be good.

Response: We thank the reviewer for the suggestion. Indeed, magnetic particles can form chains just with a static magnetic field because of the magnetic dipole-dipole interactions between the particles. However, the micro-disks used in the study tend to form hexagonal clusters under a static magnetic field, because of the capillary interaction between the disks. The angle-averaged capillary interactions are repulsive on average but when the micro-disks are stationary, the capillary torque between any two disks tends to align their sinusoidal profiles causing the disks to attract each other. The relative strength of the capillary interaction then dominates over the magnetic dipole-dipole interaction between the disks, thus causing them to form a cluster. We

would like to highlight that it is the intricate balance among the different mutual interactions between the micro-disks that enables different behaviors of the system. Eliminating any of the mutual interactions will result in a less versatile system. Moreover, using an oscillating magnetic field allows the system to form chains of varying lengths that can be controlled using the frequency of the external magnetic field.

We thank the reviewer for pointing out this confusion and have updated the ‘chain mode’ subsection of the manuscript to explain the reason for applying an oscillating magnetic field instead of a static magnetic field. The following is the modified text:

It is worth noting that although intuitively it seems that a static magnetic field could be used to create chains, this system indeed requires an oscillating magnetic field. The chain formations are formed as a result of the synchronous orientation oscillation of micro-disks throughout the collective, the flow generated by each micro-disk, and its effect on neighboring constituents. If a static magnetic field is applied, the micro-disks would cease to oscillate about their axes and the capillary interactions would dominate, which would result in formation of hexagonal clusters.

3. In the case of the high speed of the raft itself, the rafts collide with each other. Is there some consideration of physical repulsion in the simulation model needed for avoiding any overlaps between rafts? If so, the notion of the consideration would be good.

Response: We thank the reviewer for the helpful suggestion. Indeed, we use a repulsion term to prevent overlaps between the disks in simulations. This repulsion term is zero when the center-center distance (d) between two disks is greater than the diameter of each disk and it is d otherwise. We have updated the ‘Model for simulations’ subsection in the ‘Materials and Methods’ as follows:

If the center-center distance $r_{ji} < \text{lubrication threshold} (=315 \mu\text{m}, \text{ or } 1.1 R)$ and $r_{ji} \geq 300$ or $2R$,

$$\begin{aligned}
 \mu \frac{d\mathbf{r}_i}{dt} = & \sum_{j \neq i} A \left(\frac{d_{ji}}{R} \right) \left(F_{mag-on, i, j}(r_{ji}, \varphi_{ji}) + F_{cap, i, j}(r_{ji}, \varphi_{ji}) \right. \\
 & \left. + \frac{\rho \omega^2 R^7}{r_{ji}^3} \right) \hat{r}_{ji} \\
 & + \sum_{j \neq i} B \left(\frac{d_{ji}}{R} \right) F_{mag-off, i, j}(r_{ji}, \varphi_{ji}) \hat{r}_{ji} \times \hat{z} \\
 & + \sum_{j \neq i} C \left(\frac{d_{ji}}{R} \right) mB \sin(\theta - \alpha_i) \hat{r}_{ji} \times \hat{z} \\
 & + \frac{\rho \omega_i^2 R^7}{6\pi R} \left(\left(\frac{1}{d_{toLeft}^3} - \frac{1}{d_{toRight}^3} \right) \hat{x} \right. \\
 & \left. + \left(\frac{1}{d_{toBottom}^3} - \frac{1}{d_{toTop}^3} \right) \hat{y} \right), \quad i = 1, 2, \dots
 \end{aligned} \tag{7}$$

$$\begin{aligned} \mu \frac{d\alpha_i}{dt} = & G\left(\frac{d_{smallest}}{R}\right) mB \sin(\theta - \alpha_i) \\ & + \sum_{j \neq i} G\left(\frac{d_{ji}}{R}\right) T_{mag-d,i,j}(r_{ji}, \varphi_{ji}) \\ & + T_{cap,i,j}(r_{ji}, \varphi_{ji}), \quad i = 1, 2, \dots \end{aligned} \quad (8)$$

$$\mathbf{B}(t) = (B_{x_0} \cos(\Omega_x t) + B_{x_1}) \cdot \hat{x} + (B_{y_0} \cos(\Omega_y t) + B_{y_1}) \cdot \hat{y}, \quad (9)$$

where the coefficients $A(x)$, $B(x)$, $C(x)$ and $G(x)$ are lubrication coefficients.

If the center-center distance $r_{ji} < 300$ or $2R$, a repulsion term is added to the force equation,

$$\begin{aligned} \mu \frac{d\mathbf{r}_i}{dt} = & \sum_{j \neq i} A(\varepsilon) \left(F_{mag-on,i,j}(2R, \varphi_{ji}) + F_{cap,i,j}(2R, \varphi_{ji}) \right. \\ & \left. + \frac{\rho\omega^2 R^7}{r_{ji}^3} \right) \hat{r}_{ji} + \sum_{j \neq i} \frac{F_{wallRepulsion}}{6\pi R} \frac{-d_{ji}}{R} \hat{r}_{ji} \\ & + \sum_{j \neq i} B(\varepsilon) F_{mag-off,i,j}(2R, \varphi_{ji}) \hat{r}_{ji} \times \hat{z} \\ & + \sum_{j \neq i} C(\varepsilon) mB \sin(\theta - \alpha_i) \hat{r}_{ji} \times \hat{z} \\ & + \frac{\rho\omega_i^2 R^7}{6\pi R} \left(\left(\frac{1}{d_{toLeft}^3} - \frac{1}{d_{toRight}^3} \right) \hat{x} \right. \\ & \left. + \left(\frac{1}{d_{toBottom}^3} - \frac{1}{d_{toTop}^3} \right) \hat{y} \right), \quad i = 1, 2, \dots \end{aligned} \quad (10)$$

$$\begin{aligned} \mu \frac{d\alpha_i}{dt} = & G(\varepsilon) mB \sin(\theta - \alpha_i) \\ & + \sum_{j \neq i} G(\varepsilon) T_{mag-d,i,j}(2R, \varphi_{ji}) + T_{cap,i,j}(2R, \varphi_{ji}), \quad i \\ & = 1, 2, \dots \end{aligned} \quad (11)$$

$$\mathbf{B}(t) = (B_{x_0} \cos(\Omega_x t) + B_{x_1}) \cdot \hat{x} + (B_{y_0} \cos(\Omega_y t) + B_{y_1}) \cdot \hat{y}, \quad (12)$$

where ε is a small number ($10^{-10} \mu\text{m}/R$); $F_{wallRepulsion}$ is set to be 10^{-7} N.

4. In the chain motions of Fig. 4, there are some differences between the experiments and simulations. The reason for the differences should be discussed.

Response: We thank the reviewer for the suggestion. The main difference between the simulations and experiments is that the collective takes a longer time to reach steady-state in simulations as compared to the experiments. In general, the steady-state behaviors captured by the simulations for chains are in good agreement with the experiments. However, in simulating the transitions between modes, we have pushed the limits of our simulations. Such a simulation will highlight the subtle differences in the transient behavior of the system in simulations and experiments, that don't necessarily affect the steady-state behavior. Since we followed the same recipe for transitions in the simulations as in the experiments, some of the behaviors (like chains) that take longer to reach steady-state in simulations, seem to differ from the experiments. We have updated the supplementary video 3 to show the steady-state behaviors of the system in different modes.

Additionally, the curvature of the air-water interface also affects the inter-disk distance in the system. Since the boundary surrounds all sides of the collective, a concave air-water interface can drive the disks towards the center of the arena, creating an effective attraction between the disks. The simulations assume a flat air-water interface and each experiment approximates this; however, even a small concavity in the interface lowers neighbor spacing between micro-disks in the experiments, constraining the motion of individual disks. This difference is most clearly evident in the simulations of the chain mode. Measuring the curvature of the air-water interface in the experiments is extremely challenging because the curvature changes throughout a single experiment due to the evaporation of water.

The following text is added to the 'Transitions between different modes' subsection:

Additionally, a concave air-water interface can drive the disks towards the center of the arena, creating an effective attraction between the disks. The simulations assume a flat air-water interface and each experiment approximates this; however, even a small concavity in the interface lowers neighbor spacing between micro-disks in the experiments, constraining the motion of individual disks. This difference is most clearly evident in the simulations of the chain mode.

5. In Fig. 5 a (ii), the magnetic field gradient's unit is G/mm^2 . Please double-check the unit of the field gradient.

Response: We thank the reviewer for pointing out the mistake in units. We have updated the figure so that the units for the magnetic field gradient reads G/mm . The updated fig. 5 is added below:

Fig. 5. Magnetic gradient-assisted collective navigation through confined environments and contact-based object transport. (a) Locomotion speed comparison for the rotation, static, oscillation, and chain modes with $N = 126$ micro-disks driven at different frequencies when the gradient is $F = 0.7$ Gauss / mm. The chains move faster than other modes. The error bars represent standard deviations over 5 s. (b) Displacement of each mode over time when Hz. The chains move faster and get slowed down due to the physical boundary around 2.5 s. This contributes to large standard deviations in chain speed in (a). (c) Representative images showing the trajectories of static (left), oscillating (middle left), rotating (middle right), and chain (right) collectives under the influence of a magnetic field gradient along the y axis. (d) Y chain of seven micro-disks driven with magnetic field gradients to produce MPI and C trajectories (Supplementary Movie 7). (e) 17 micro-disks transition between different modes to navigate through narrow passages (Supplementary Movie 8). (f) The collective switches between the modes static, X chains, and Y chains to locomote and push an object. The brightness of the images in (c-f) is enhanced using photoshop for better visualization. The symbols on the sub-images at the bottom, in (c), and top, in (e-f), represent the mode the collective exhibits. These symbols correspond to those shown in Fig. 1a. The sub-images in (f) are labeled at the bottom with the function that the collective performs.

6. The coil system set up in the paper generates some tens of mT magnetic fields in about a hundred frequencies. Some confirmation of the capability of the generation of dynamic magnetic fields of the setup e.g. bode plot of current in the coil system would improve the paper..

Response: We thank the reviewer for this helpful suggestion. We have added the bode plot for the current in the coil system and the comparison of the frequency of the output and input signal in the supplementary fig. 1. The bode plot for magnitude drops below the -3dB line after 250 Hz. The updated part of the supplementary fig. 1 is added below:

Supplementary Fig. 1. Experimental setup. (a) Two-axis Helmholtz coils and imaging setup: (1) Manual and computer view onto stage; (2) X-coils; (3) Y-coils; (4) Stage. (b-g) Arena designs for experimental demonstrations: (b) Maze arena. (c) Ball Motion Arena. (d) Quadrotor object. (e) C-shape object. (f) Rod-like object. (g) Star-like object. (h) Plot of frequency of output coil current vs input signal to generate the magnetic field. The output current has the same frequency as the input signal. (i) Bode plot of magnitude and phase of the coil current. The magnitude of the output current falls below the 3 dB line after 200 Hz.

Reviewer #3:

This manuscript entitled ‘Microrobot Collectives with Reconfigurable Morphologies, Behaviors, and Functions’ presents a collective microrobot system that shows different behaviors under actuation of externally magnetic field. Meanwhile, the inter-mode transitions and various tunability of the swarm are also demonstrated by experiments and simulations. As we all know, the microrobot swarms show the advantage of faster speed, higher carrying efficiency and group deformation, which could overcome the disadvantage of low delivery efficiency and slow speed of a single microrobot. Moreover, microrobot swarm should be composed of multiple microrobots with motion behavior. However, the motion behavior of a single micro-disk under different magnetic fields was not exhibited. Then, the control modes and six motion forms under different magnetic fields in the manuscript do not show any new ideas, highly similar with the theme of the previous publications (Nature Communication (2019) 10:5631; Sci. Robot. 4, eaav8006 (2019)). Above all, in terms of application, transporting and rotating objects shown by the microrobot swarm are not bright spots. Hence, I cannot recommend the publication of this manuscript.

Response: We thank the reviewer for this comment and appreciate the reference to the two previous publications. However, our current study has major contributions and differences with compared to these previous works in below aspects:

1. We introduce six collective modes that can be switched between on-demand. To the best of our knowledge, our system exhibits **more distinct collective behaviors** than any other study in the literature. Each different mode can be reached by tuning two perpendicular oscillating magnetic fields. We characterize the outstanding features of each mode as a function of different frequencies and numbers of micro-disks in the collective.
2. **Two novel modes** presented are specific to our system and have **no counterpart** in the literature:
 - a. The first novel mode is the static mode which enables the collective to remain fixed even though each micro-disk is oscillating about its own center of mass.
 - b. The second novel mode is the Gas-like Self-Propelling Pairs (GaSPP) mode, which enables the collective to translate in groups of two and occupy a given space.
3. To the best of our knowledge, our system is the **first to demonstrate** a transition between globally driven behaviors to a mutual interactions-dominated self-propulsion behavior. Such transition allows our systems to be useful for both collective robotics and fundamental studies.
4. Our system enables the use of **mechanical coupling** at the millimeter scale between the micro-disks and surrounding passive particles and boundaries. We demonstrate that a collective can alter the surrounding medium itself to transport and rotate surrounding particles **for the first time**; this is achieved through the azimuthal flow field enabled by the micro-disks’ actuation.
5. We present simulations that replicate each of the collective mode behaviors and the reconfigurability between the different collective behaviors.

To highlight the major differences between our work and the mentioned papers more clearly, we have now modified the following lines in the Introduction section:

Other microrobotic systems have demonstrated ribbon, chain or vortex formations^{25–27,46–48}, locomotion^{41,49}, and object manipulation^{50,51}. One study shows locomotion of two microrobotic swarms that can navigate through various mediums and complex environments; however, each swarm demonstrates a single mode, one creates ribbon-like formations while the other creates vortex-like formations⁵². Another experimental system produces four emergent modes (liquid, chains, vortices, and ribbons)²⁶, and it relies on a solid substrate for symmetry breaking to enable the formations and locomotion. Although this system exhibits four modes, the relative strength of different mutual interactions in the system is not clearly tunable; this limits the control of the collective's size, inter-particle separation, and each mode's local order.

We have also updated the **Supplementary Video 1** by adding the behavior of a single micro-disk when different magnetic field profiles are used. The video clearly shows that a single micro-disk does not translate for any of the magnetic signal. It just oscillates in place, to follow the direction of the external magnetic field. This video highlights the importance of the mutual interactions between the micro-disks to generate the different behaviors.

REVIEWERS' COMMENTS

Reviewer #1 (Remarks to the Author):

Sincere thanks for the substantial updates made to the manuscript. All my concerns have now been addressed. Overall this is a very nice paper.

Reviewer #2 (Remarks to the Author):

I carefully checked the response and revised manuscript.
My Concerns have been considered and addressed properly
Now the manuscript can be considered for publication

Reviewer #3 (Remarks to the Author):

The revised version has indeed improved the quality of this manuscript. However, they donot answer my concern on the innovation of this manuscript. as mentioned before, too many papers on magnetic microrobot or micromotor swarms have been published last three years (e.g. Tierno group). In my opinion, this manuscript is basically a collective of previous reports. I donot find any new scientific contribution. Frankly, I antipate they may focus on the potential applicaiton of these magnetic microbots or introduce more intelligent controls.

Response to reviewers' comments: We thank the reviewers for their helpful and constructive comments on our manuscript. Below is our response to each reviewer's comments. The original comments are in italic, and our response are indented and in normal text in dark blue color. The new data and edits in the revised manuscript and SI are highlighted in yellow.

Reviewer #1 (Remarks to the Author):

Sincere thanks for the substantial updates made to the manuscript. All my concerns have now been addressed. Overall this is a very nice paper.

We thank the reviewer for their positive assessment of our work. We are grateful for all the input from the reviewer throughout the review process.

Reviewer #2 (Remarks to the Author):

I carefully checked the response and revised manuscript.

My Concerns have been considered and addressed properly

Now the manuscript can be considered for publication

We thank the reviewer for the positive assessment of our work and their thoughtful suggestions throughout the review process.

Reviewer #3 (Remarks to the Author):

The revised version has indeed improved the quality of this manuscript. However, they donot answer my concern on the innovation of this manuscript. as mentioned before, too many papers on magnetic microrobot or micromotor swarms have been published last three years (e.g. Tierno group). In my opinion, this manuscript is basically a collective of previous reports. I donot find any new scientific contribution. Frankly, I antipate they may focus on the potential applicaiton of these magnetic microbots or introduce more intelligent controls.

We appreciate the reviewer's comments and suggestions. We have specified our particular contributions in the main text and have also discussed the future extensions of this work and its applications (cited below). We agree with the reviewer that implementing intelligent control of the behaviors is an interesting study in its own right, and it will be part of the future study for this microrobot collective; however, we think that the novel behaviors and the on-demand reconfiguration among the behaviors to complete different tasks are significantly different from previous reports, and they as a whole provide the type of scientific contribution that will inspire a wide variety of future microrobot collective studies. Below we quote part of the main text that stress the novelty and the unique contributions of this study.

Lines 75-82:

“An ideal system should enable an external control parameter, like the magnetic field frequency, to dynamically program relative dominance among different particle-particle and particle-environment interactions to reconfigure between several tunable modes and their functions. Moreover, a system that can transition on-demand from globally driven behaviors to a mutual interactions - dominated self-propelling behavior (like self-propelling Janus particles^{30,31}) is yet to be realized. Such a system would not only be useful for robotics applications but also for fundamental studies to explore the link between globally driven and active systems.”

Lines 88-99:

“Each behavior is enabled by several particle-particle and particle-fluid interactions (e.g., hydrodynamic, capillary, and magnetic dipole-dipole interactions) that are controlled by external magnetic fields. Some of these behaviors are unique to our system and have no counterpart in other systems. Moreover, our system shows both isotropic (rotating collectives) and anisotropic (chains) behaviors, and it also transitions from such globally driven behaviors to mutual interactions - dominated self-propelling behavior (like self-propelling Janus particles). The system's versatility and the micro-disks' ability to remain at the fluid-air interface is well-suited for practical applications like manipulating cells within a microfluidic chip, guiding development of micromachines, acting as a model system for exploring self-organization in soft matter, and studying collective behaviors that could finally translate to three dimensions and be used for microscale packaging²² and medical applications, such as targeted active drug or other cargo delivery⁵⁴⁻⁵⁶.”

Lines 269-282:

“The GaSPP mode opens up possibilities for several fundamental studies investigating the systems of active particles. For example, studies like those investigating the emergence of

directed motions in a collective of active particles⁵⁷, could also be possible using the GaSPP mode, by tuning the density of the micro-disks in GaSPP mode (the speed of the pairs can be tuned using the external magnetic field) to study the emergence phenomenon in such systems. The GaSPP mode could be advantageous over other systems, like the quincke rollers, or the active janus particles, because the mutual interactions among the micro-disks in GaSPP mode are relatively simpler than the non-trivial fluidic interactions among the quincke rollers and also GaSPP does not require any fuel (as is the case for the chemotactic janus particles). Moreover, the speed of the pairs in the GaSPP can be tuned by the external magnetic field. The GaSPP behavior could also be useful for demonstrating theories such as the long-range order in active systems. The GaSPP mode can be useful for robotics studies as well. For example, the GaSPP can be used as the microrobot swarm to create superstructures like the ones developed using the rodlike vibrating robots^{58,59}.”

Lines 362-364:

“The complex interplay between parameters like magnetic field gradient, boundary conditions, and bi-axial magnetic field frequencies opens up many avenues for researchers to explore optimal control methods for transporting a collective within complex and/or dynamic environments.”

Lines 486-499:

“Our system is versatile and demonstrates several useful behaviors and functions at this scale and at the fluid-air interface. In addition, two of the presented behaviors are novel and unique to our system, i.e., the static collective composed of dynamic micro-disks and the gas-like formation composed of self-propelling pairs. Unlike any other system in the literature, our system transitions from globally driven to mutual interactions-dominated self-propelling behavior. Moreover, our system enables the use of traditional mechanical coupling at the millimeter scale for torque transfer, paving a way for the transfer of knowledge of traditional mechanical systems to build machines at smaller length scales. One feature of our system is that even an incredibly small number of micro-disks, as small as 7, can exhibit all the formations demonstrated in this study. This capability could be very advantageous in scenarios when the collective needs to be broken into smaller groups while still retaining the ability to reconfigure. We envision that our system would be useful in performing several tasks applicable to microscale self-assembly and packaging²², and its use as reconfigurable micromachines in biomedical and environmental applications.”